# From Psychostasis to the Discovery of Cardiac Nerves: The Origins of the Modern Cardiac Neuromodulation Concept

**DOI:** 10.3390/biology13040266

**Published:** 2024-04-16

**Authors:** Beatrice Paradiso, Dainius H. Pauza, Clara Limback, Giulia Ottaviani, Gaetano Thiene

**Affiliations:** 1Lino Rossi Research Center, Department of Biomedical, Surgical and Dental Sciences, Faculty of Medicine and Surgery, University of Milan, 20122 Milan, Italy; giulia.ottaviani@unimi.it; 2Consultant Cyto/Histopathologist (Anatomic Pathologist) Anatomic Pathology Unit, Dolo Hospital Venice, 30031 Dolo, Italy; 3Faculty of Medicine, Institute of Anatomy, Lithuanian University of Health Sciences Kaunas, 44307 Kaunas, Lithuania; dainius.pauza@lsmu.lt; 4Oxford University Hospitals, NHS Trust, Oxford OX3 7JH, UK; clara.limbaeck@ouh.nhs.uk; 5Department of Biomedical, Surgical and Dental Sciences, Faculty of Medicine and Surgery, University of Milan, 20122 Milan, Italy; 6Department of Pathology and Laboratory Medicine, McGovern Medical School, University of Texas Health Science Center at Houston, Houston, TX 77054, USA; 7Cardiovascular Pathology, Department of Cardiac, Thoracic, Vascular Sciences and Public Health, University of Padua, 35122 Padua, Italy; gaetano.thiene@unipd.it

**Keywords:** psychostasis, autonomic nervous system, cardiac innervations, cardiac neuromodulation, polyvagal theory, long COVID

## Abstract

**Simple Summary:**

The review aims to explore the historical development of cardiac innervations and neuromodulation, tracing its origins back to the ancient Egyptian concept of “psychostasis”—the ritual weighing of the heart to determine one’s fate in the afterlife. This ancient belief in the heart as the center of human wisdom, emotions, and memory laid the foundation for the later advancements in the scientific understanding of cardiac innervation. Importantly, this ancient cardiocentric worldview also established the ancestral ethical foundations for the study and application of cardiac neuromodulation. The Egyptians believed that the “lightness of the heart”—achieved through a balance of good and bad deeds—was crucial for one’s wellbeing and fate in the afterlife. This ethical framework parallels the modern understanding of the importance of balanced cardiac autonomic control, mediated by the vagus nerve, for overall health and social engagement. The review chronicles key milestones, from Aristotle and Galen’s early physiological studies, to the pioneering anatomical work of Leonardo da Vinci, Vesalius, and William Harvey’s discovery of the circulatory system. It highlights the contributions of 17th century researchers like Richard Lower, who demonstrated the heart’s neural control, and Albrecht von Haller, who discovered the heart’s automaticity. The review emphasizes how the study of cardiac innervation and neuromodulation has regained prominence, particularly in understanding the cardiorespiratory symptoms and dysregulation observed in post-acute sequelae of SARS-CoV-2 infection (long COVID) and other acute respiratory infections. It draws parallels between the ancient concept of “weighing the heart” in psychostasis and the modern understanding of the “balance of different hormonal, chemical, electrical, and contrasting nervous stimuli” in cardiac neuromodulation. By tracing this historical evolution, the review aims to bridge the gap between ancient concepts like psychostasis and the modern applications of cardiac neuromodulation for cardiovascular health and disease management. Understanding the complex interplay between cardiac function, neural modulation, and overall wellbeing is crucial for addressing long-term cardiorespiratory issues, akin to the ancient notion of achieving “lightness of the heart”. The review also highlights the relevance of the polyvagal theory, which emphasizes the role of the vagus nerve in regulating the heart’s function and its connection to emotional and social engagement. This provides an ethical and physiological framework for understanding the importance of cardiac neuromodulation in promoting overall health and well-being.

**Abstract:**

This review explores the historical development of cardiology knowledge, from ancient Egyptian psychostasis to the modern comprehension of cardiac neuromodulation. In ancient Egyptian religion, psychostasis was the ceremony in which the deceased was judged before gaining access to the afterlife. This ritual was also known as the “weighing of the heart” or “weighing of the soul”. The Egyptians believed that the heart, not the brain, was the seat of human wisdom, emotions, and memory. They were the first to recognize the cardiocentric nature of the body, identifying the heart as the center of the circulatory system. Aristotle (fourth century BC) considered the importance of the heart in human physiology in his philosophical analyses. For Galen (third century AD), the heart muscle was the site of the vital spirit, which regulated body temperature. Cardiology knowledge advanced significantly in the 15th century, coinciding with Leonardo da Vinci and Vesalius’s pioneering anatomical and physiological studies. It was William Harvey, in the 17th century, who introduced the concept of cardiac circulation. Servet’s research and Marcello Malpighi’s discovery of arterioles and capillaries provided a more detailed understanding of circulation. Richard Lower emerged as the foremost pioneer of experimental cardiology in the late 17th century. He demonstrated the heart’s neural control by tying off the vagus nerve. In 1753, Albrecht von Haller, a professor at Göttingen, was the first to discover the heart’s automaticity and the excitation of muscle fibers. Towards the end of the 18th century, Antonio Scarpa challenged the theories of Albrecht von Haller and Johann Bernhard Jacob Behrends, who maintained that the myocardium possessed its own “irritability”, on which the heartbeat depended, and was independent of neuronal sensitivity. Instead, Scarpa argued that the heart required innervation to maintain life, refuting Galenic notions. In contemporary times, the study of cardiac innervation has regained prominence, particularly in understanding the post-acute sequelae of SARS-CoV-2 (Severe Acute Respiratory Syndrome Coronavirus 2) infection (PASC), which frequently involves cardiorespiratory symptoms and dysregulation of the intrinsic cardiac innervation. Recently, it has been recognized that post-acute sequelae of acute respiratory infections (ARIs) due to other pathogens can also be a cause of long-term vegetative and somatic symptoms. Understanding cardiac innervation and modulation can help to recognize and treat long COVID and long non-COVID-19 (coronavirus disease 2019) ARIs. This analysis explores the historical foundations of cardiac neuromodulation and its contemporary relevance. By focusing on this concept, we aim to bridge the gap between historical understanding and modern applications. This will illuminate the complex interplay between cardiac function, neural modulation, cardiovascular health, and disease management in the context of long-term cardiorespiratory symptoms and dysregulation of intrinsic cardiac innervations.

## 1. The Heart and Psychostasis in Ancient Egypt and in Early Christianity

Cardiac neuromodulation represents the intricate physiological control exerted on the heart through a diverse array of endogenous and exogenous mechanisms, aimed at modulating nerve activity within the cardiovascular system. This field has a long history, dating back to ancient Egyptian psychostasis, which involved the use of a scale to weigh the heart after death to determine a person’s fate in the afterlife. The concept of psychostasis is based on the idea of balancing good and bad deeds, as represented by the weighing of the heart against a feather. In this review, we will explore the historical development of cardiac neuromodulation, drawing inspiration from the ancient Egyptian concept of psychostasis. We will describe how the balance of different hormonal, chemical, electrical, and contrasting nervous stimuli is akin to the weighing of the heart in psychostasis. By examining the historical evolution of cardiology knowledge from ancient Egyptian psychostasis to the contemporary understanding of cardiac neuromodulation, we aim to shed light on the potential benefits and challenges of these approaches for the treatment of cardiovascular diseases. We will begin by discussing the historical development of cardiac neuromodulation, tracing its roots from ancient Egyptian psychostasis to the discovery of cardiac nerves in the 17th century. We will then examine the mechanisms and therapies of cardiac neuromodulation, highlighting its potential applications in the treatment of various cardiovascular diseases. Finally, we will discuss the relevance of cardiac neuromodulation to psychostasis, drawing parallels between the balance of good and bad in psychostasis and the balance of different stimuli in cardiac neuromodulation.

The heart was revered in ancient Egypt as the seat of intelligence, religion, and spirituality, as well as the organic engine of the body. It was regarded as one of the eight essential bodily parts. Unlike other organs, the heart needed to be carefully conserved in the mummy to ensure its immortality. Early in the first dynasty, the heart was depicted as a hieroglyph connected to eight vessels [1,2].

30 centuries ago, Egyptian physicians established an original theory of cardiovascular physiology that has endured for 30 centuries [3]. *The Treatise on the Heart*, the first cardiology book in human history, is one of the nine sections of the *Ebers Papyrus* (1534 BC) [4]. Unfortunately, the first translations of this papyrus were very inexact, which obscured awareness of the remarkable accomplishments of medicine, particularly of Pharaonic cardiology. Thanks to more precise translations, particularly that of Thierry Bardinet [5], we can now recognize that the Egyptians were the real pioneers of medicine as well as cardiology.

Bardinet elucidated the concept of the heart and its “components” in ancient Egypt based on graphic representations and the interpretation of hieroglyphs that represent the heart. The Egyptians, in fact, combined three distinct concepts under this term, both complementary and intricate (Figure 1(IA)).

The first, termed heart-haty (Figure 1(IA)), corresponded to the anatomical heart responsible for the circulation of fluids throughout the human body.

The second, defined as heart-ib or, according to Bardinet’s interpretation, interior-ib (Figure 1(IB)), grouped the organism’s structures responsible for the whole vital function, with the exception of the heart-haty [5].

The third, finally, symbolized the center of thought, intelligence, and memory, responsible for collecting all information from the senses, anticipating the concept of cardiac innervation.

However, the ancient Egyptians were unable to distinguish between arteries, veins, tendons, and nerves, which were all referred to as “metw”, i.e., network texture, supporting the cardiocentric doctrine (Figure 1II). Their understanding of the heart’s structure was primarily influenced by their spiritual and religious convictions, but records demonstrate that they could link the pulse to cardiac contractions (Figure 1III). The term heart-haty itself means “the one who stands in front, the one who commands” [4,6].

Despite the anatomical accuracy of the various hieroglyphs representing the heart, particularly the representation of eight vessels alluding to the pulmonary artery, aorta, superior and inferior vena cava including the four pulmonary veins (Figure 1II), which dates back to the first dynasty (3000 BC), no other civilization contemporaneous to Pharaonic Egypt depicted the heart with such precision. This suggests that only a scholar with access to heart anatomy could have designed such hieroglyphs.

The third “constituent” of the heart, the “heart-soul”, embodies the concept of individual and social value, particularly in the context of the ancient Egyptian religion’s ceremony known as “psychostasis”. This ceremony, described in chapter 125 of the *Book of the Dead* (*Papyrus of Ani*, 1275 BC) [7] (Figure 1IV), involved the deceased being weighed against the feather of Maat, the goddess of order, truth, and righteousness. To the Egyptians, the heart, not the brain, was the source of human discernment, emotions, and memory. In the weighing ritual, the heart of the deceased was placed on a scale opposite the feather of Maat, and their fate was determined by whether the heart balanced the feather or weighed heavier (indicating a life of sin).

Although psychostasis was a fundamental and ubiquitous aspect of Egyptian thought, the earliest Egyptian literature on the afterlife, dating back to the Old Kingdom (2425–2300 BC), did not mention it. Instead, postmortem judgment was depicted using worldly courtroom symbolism and metaphors. This changed during the First Intermediate period (2200–2050 BC) with the introduction of the “Instruction for King Merikare”, which introduced the concept of measuring the deceased’s good and bad deeds in two separate piles. According to Samuel Brandon, this new element of court procedures was developed in response to Egyptians’ distrust of earthly justice [8,9]. By weighing good deeds against bad deeds, the afterlife judgment became more objective and impartial, ensuring that individuals’ destinies after death were based on their own actions, not on the biases of a divine judge. This concept of weighing was further developed in the Coffin Texts of the Middle Kingdom, and it was fully incorporated into Egyptian mythology during the New Kingdom (1580–1090 BC), with the Book of the Dead describing the weighing of the deceased’s heart against the feather of Maat on a set of scales.

The ancient Egyptians believed that the heart was a sentient entity that acted independently, sometimes even against its owner. Some Egyptians even personified the heart as a tiny deity residing within human beings. The heart was considered the repository of a person’s memories and intelligence, allowing them to retain both positive and negative recollections of their earthly life. Moreover, the heart represented the essence of an individual’s moral and ethical compass.

**Figure 1 biology-13-00266-f001:**
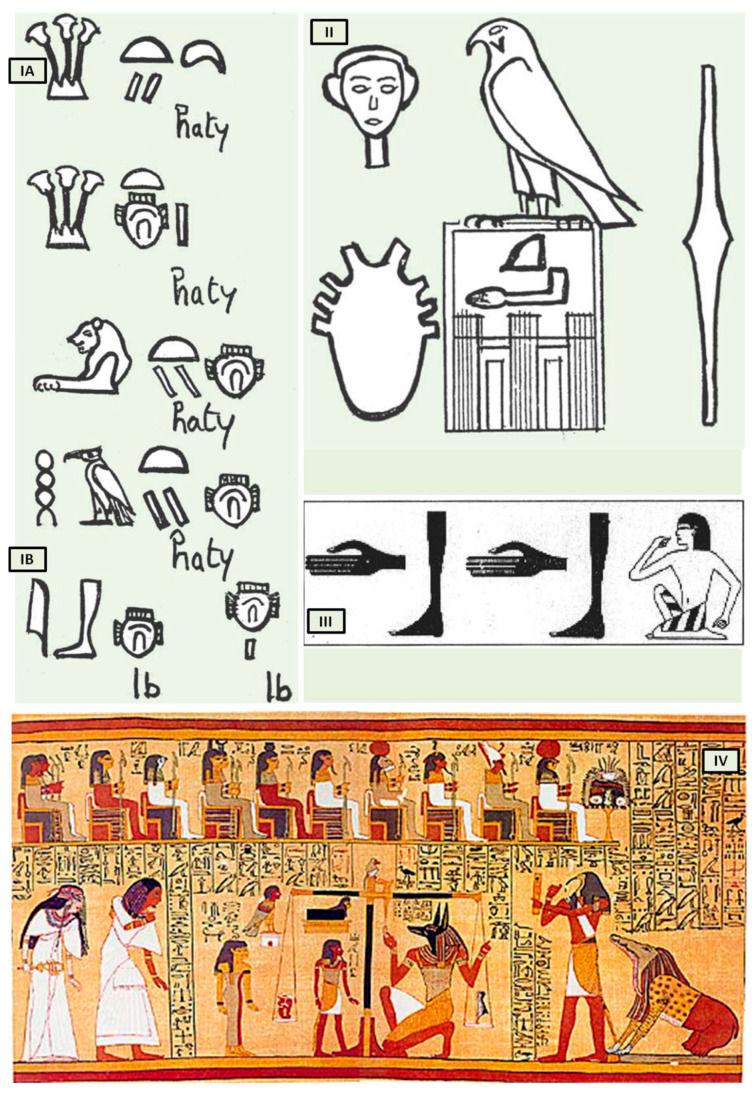
The heart in ancient Egypt. (**I**) The heart and the interior-ib in medical texts. (**A**) Heart-haty. (**B**) Heart-ib or interior-ib [adapted from B. Ziskind [1]]; (**II**) representation of the heart in the titling of the pharaoh. All the names brought by the king of Egypt constitute his title. Title of Horus Qâ, pharaoh of the 1st dynasty [adapted from B. Ziskind [1]]; (**III**) the “debdeb” hieroglyph; pulsate in the *Ebers Papyrus* [adapted from B. Ziskind [2]]; (**IV**) Anubis weighing the heart of Ani. [adapted from *Papyrus of Ani*, 1250 b.C, during the Nineteenth Dynasty of the New Kingdom of ancient Egypt. Egyptians dealing in illegal antiques in Luxor in 1888 were responsible for the discovery of the scroll. The purchase was made by E. A. Wallis Budge and, finally, by the British Museum [7]].

In Egyptian mythology, the goddess Maat, daughter of the sun god Ra, symbolized Egyptian ethics and cosmology. She embodied the Egyptian concept of cosmic and social order, while Ra himself embodied the ultimate cosmic order. Maat was regarded as the embodiment of “truth”, “justice”, “rectitude”, “equilibrium”, “cosmic law”, or “order” due to her association with social harmony. Essentially, Maat represented a standard or benchmark against which a person’s character and actions could be judged in this world [10].

The concept of psychostasia originated in Egypt and spread to various cultures and religions [10,11]. In the practice of psychostasia, the ancient Greeks, Muslims, Hindus, and Buddhists, unlike the Egyptians, who weighed the heart, placed the entire body on the scales as a symbolic process to weigh the soul, fate, and actions of the individual. This symbolic act aimed to evaluate the soul based on one’s deeds and inner qualities rather than focusing on specific body parts.

The psychostasia of ancient Greece is depicted in the Iliad attributed to Homer, likely written in the late 8th or early 7th century BC. This poem uses the idea of “weighing of the souls” for judgment, with Zeus weighing the fates of Achilles and Hector (Il. 22. 209 ff., [12]). The Greek adaptation of psychostasia differs from the Egyptian version by focusing on individual fates rather than moral worth, reflecting a pre-mortem judgment [13,14]. This adaptation illustrates how ancient literature tailored concepts to fit their beliefs. While specifics vary, the core purpose remains consistent: an impartial method for divine figures to determine judgment. This practice is common in many religions like Islam, Zoroastrianism, Hinduism, Buddhism, and medieval Christianity, describing post-mortem divine judgment using psychostasia imagery. In Islam, the Quran was revealed to Prophet Muhammad in the 7th century CE through the archangel Gabriel as guidance for humanity. While the Greeks weighed the fates or lives of warriors (Keres or psychai), the Quran [15] focuses on weighing good deeds. Despite this difference, both traditions use scales for judgment, influencing one’s destiny based on what is weighed. Quranic verses like 7:8–9 and 23:102–103 mention psychostasia [15], aligning with the ancient Greek concept of weighing souls.

In Christian depictions of the Last Judgment, Archangel Michael weighs souls on a scale, mirroring the Egyptian concept of Maat overseeing the weighing of hearts (Figure 2).

While the Old Testament (3rd century BC) does not explicitly mention Michael weighing souls on a scale, it does metaphorically portray God assessing people’s attitudes and actions using the image of a scale of justice (Proverbs 16:11). The Testament of Abraham, a pseudepigraphic book of the Old Testament, narrates Abraham’s journey to heaven, guided by Michael. During this journey, they encounter an angel holding a scale for weighing souls. This text, likely written in the 2nd century AD, suggests that the concept of psychostasis had already permeated Jewish beliefs by this time.

**Figure 2 biology-13-00266-f002:**
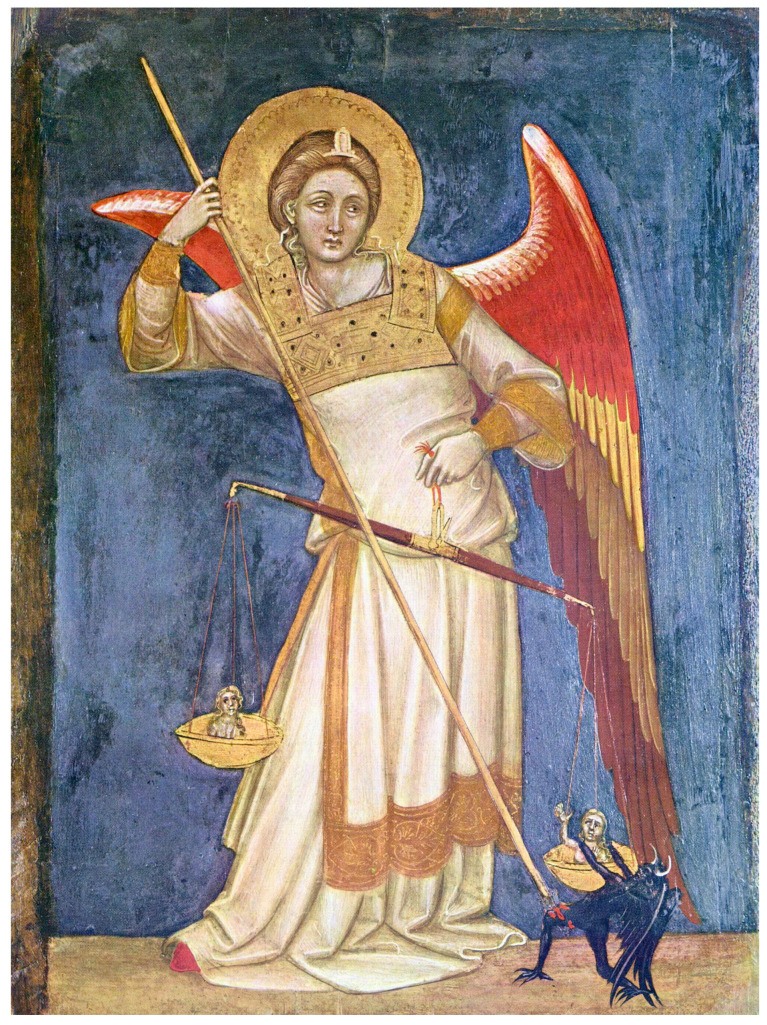
Archangel Michael weighs souls and defeats Satan [adapted from Guariento di Arpo (1310–1370 AD)—Padua, Museo Civico].

In the New Testament (I century AD), the concept of psychostasis reappears in the original context of heart weighing: ^34^“Be careful, or your hearts will be weighed down with carousing, drunkenness, and the anxieties of life, and that day will close on you suddenly like a trap.” [16].

According to Christianity, humanity has to live paying attention to its own heart, first of all, because it is the house of life and the door of God. The incarnation is not over; it happens continuously. In the Eucharist, we find the flesh of Jesus.

And again, in the New Testament, we read: ^28^“Come to me, all you who are weary and burdened, and I will give you rest. ^29^Take my yoke upon you and learn from me, for I am gentle and humble in heart, and you will find rest for your souls. ^30^For my yoke is easy and my burden is light.” [17].

In Christianity, the weighing of the heart against the feather of Maat, Egyptian, which had a role of social punitive coercion (do not do evil because it damages society and you will be punished), is replaced with the weighing of one’s heart against the lightness of the incarnate sacramental bread of Christ. Jesus suffered in defense of the last and the oppressed; his heart is light because he offers mercy (from the Latin misericors and from misereor (I have pity) and cor (heart)). “Cor” of the Christian Divinity is an empathic heart. Nowadays, we could say that it is a “vagal” heart. That is, it has a calming effect on the heart itself, reduces sympathetic responsiveness, and promotes social engagement behaviors.

In this regard, all Catholic communities in the world have handed down the story of Eucharistic miracles [Blessed Carlo Acutis planned and organized The Eucharistic Miracles of the World, an international exhibition. http://www.miracolieucaristici.org (accessed on 6 April 2022) [18]].

The Catholic faith does not hinge on Eucharistic miracles, and Christians are not obligated to believe in any private revelation, even those authorized by the Church. However, it is intriguing to revisit the Eucharistic miracle that occurred between 730 and 750 in Lanciano, Italy [19], within the context of our analysis. According to historical tradition, a monk who harbored doubts about the genuine presence of Christ in the Eucharist witnessed the bread and wine transforming into flesh and blood during the Mass when he pronounced the words of consecration (Figure 3I) [19].

Numerous studies have attempted to analyze the relics, with varying outcomes influenced by scientific advancements over the centuries.

Professor Odoardo Linoli, a specialist in anatomy, histology, chemistry, and clinical microscopy and former head of the Pathology Laboratory at Arezzo’s “Spedali Riuniti” Hospital, conducted an examination of the specimens in 1971. He documented his findings in Quaderni Sclavo di Diagnostica Clinica e di Laboratori, which was indexed in PubMed [20]. In 1981, Ruggero Bertelli, a retired professor of human anatomy at the University of Siena, corroborated Linoli’s analysis.

The reports of the investigations into the ancient flesh of Lanciano conducted in 1971 and 1981 indicate that it is striated muscle tissue that appears to be of cardiac origin owing to its extensive syncytial connections between the fibers, as well as arterial and venous vessels and two slender branches of the vagus nerve [19].

While it undoubtedly does not confirm that the Lanciano host of the 8th century is the original one transformed into myocardium, this remaining an exclusively faith-based matter, it nonetheless offers us an anthropological and historical perspective from which to approach the study of cardiac innervation through the evolution of the psychostasis concept. Strikingly, the histological features of the Lanciano host’s flesh closely resemble, from a neuroanatomical standpoint as well, the previously proposed description of psychostasis in paleo-Christianity, particularly the “vagal heart” of Jesus (Figure 3II,III) [19].

**Figure 3 biology-13-00266-f003:**
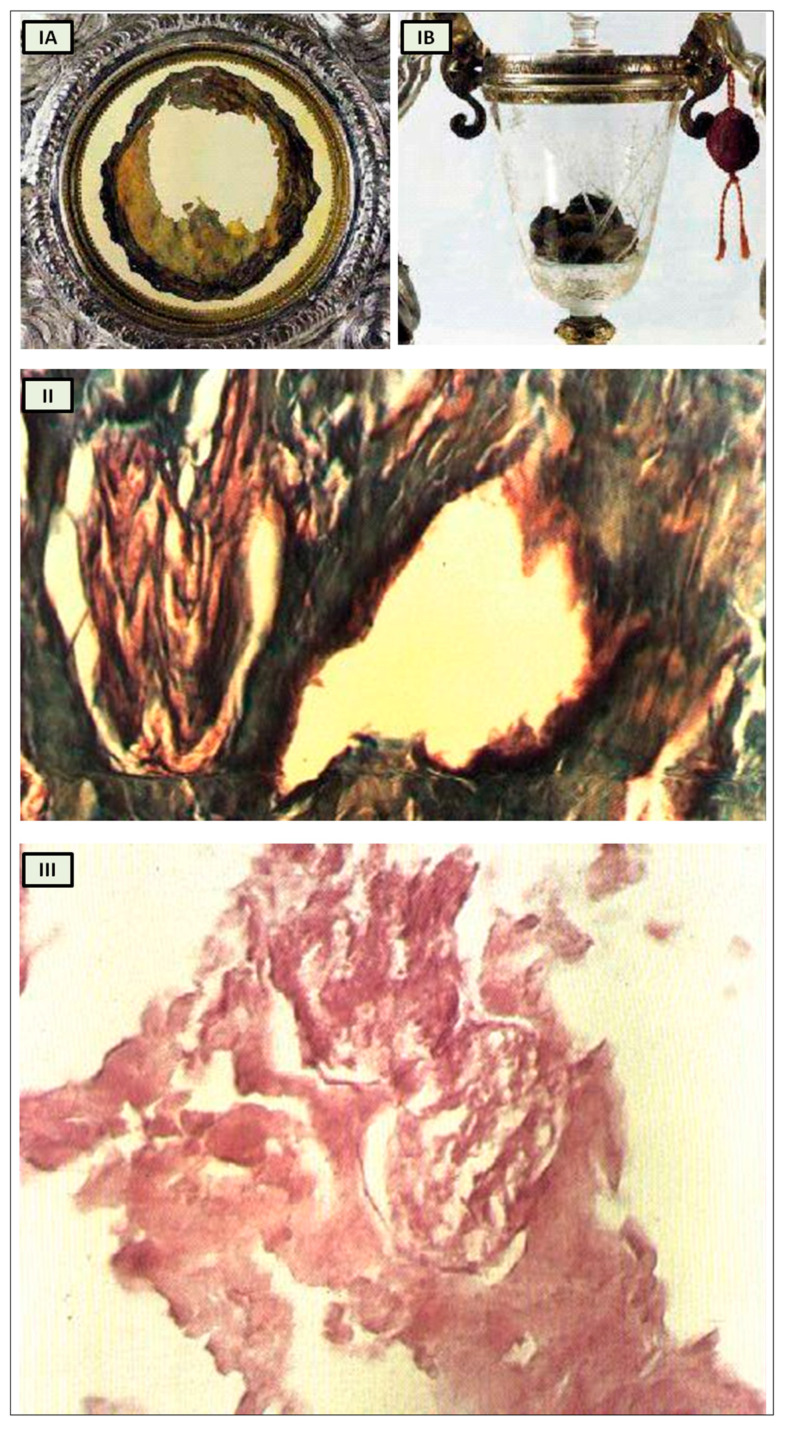
Miracle Heart in Lanciano, Italy. (**I**) The relics of Lanciano’s Eucharistic miracle. When an 8th-century priest questioned Christ’s presence in the Eucharist, the host was turned into flesh (**A**) and the wine into blood (**B**); (**II**) Miracle Heart in Lanciano. Mallory × 250. A branch of the vagal nerve and an artery; (**III**) Miracle Heart in Lanciano. Eosine multiplied × 350. A vagal nerve branch. The perineurium is narrow, and the fascicular structure is well preserved. [adapted from N. Nasuti [19]].

In modern times, we can inquire into the scientific basis of anima-bios well-being as reflected in the concept of a “light heart”. Does it hold physiological significance to say that an individual approaches life with a “light heart”? Which autonomic nervous system (ANS) component could be activated in this sensation?

Conversely, what factors contribute to feelings of fatigue, a “heavy heart”, and depression, common symptoms associated with stress and pathologies prevalent in the industrial age?

Building on these considerations, let us examine the symptoms and sequelae of a novel respiratory infection that emerged in December 2019 in Wuhan, China.

This newly identified coronavirus, originating from animals (zoonotic), spread rapidly around the world. In response, the World Health Organization (WHO) designated it as 2019-nCoV, signifying a novel coronavirus from 2019. The disease caused by this virus was later named COVID-19, and on 11 March 2020, WHO declared it a global pandemic.

COVID-19 is responsible for severe acute respiratory syndrome coronavirus 2 (SARS-CoV-2), manifesting as atypical pneumonia alongside flu-like respiratory symptoms. This infection poses a significant global health threat, leading to post-acute sequelae of COVID-19 (PASC) in affected individuals. Studies indicate that 30% of COVID-19 patients experience persistent symptoms for up to 9 months post-infection. Those recovering from COVID-19 may endure prolonged symptoms categorized as long COVID or post-COVID syndrome [21,22].

Long COVID presents various adverse outcomes, including cardiovascular, thrombotic, and cerebrovascular diseases, type 2 diabetes, myalgic encephalomyelitis/chronic fatigue syndrome (ME/CFS), postural orthostatic tachycardia syndrome, and other dysautonomic events. Effective treatments for these debilitating symptoms are yet to be validated, with some conditions expected to persist for years or even a lifetime. It remains unclear whether long COVID is a different disease entity with unclear pathophysiology or a spectrum of prolonged viral infection.

Respiratory distress observed in COVID-19 and long COVID patients is not solely attributed to atypical pneumonia. Research suggests that aberrant ventilatory responses may stem from direct brainstem involvement and para-infective mechanisms affecting the peripheral nervous system [23,24].

In particular, the study by Jareonsettasin et al. explores the intriguing phenomenon of impaired perception of dyspnea to tachypnea in COVID-19 patients, shedding light on the underlying mechanisms driving this inappropriate ventilatory response. The research uncovers the role of increased afferent feedback from chest wall mechanoreceptors and muscle stretch receptors, highlighting the abnormality where a significant portion of patients exhibit a reduced dyspneic response to tachypnoea. This anomaly is attributed to factors such as maintained neuromechanical coupling and potential interruptions in central processes when comparing the expected and actual consequences of breathing motor commands. Furthermore, the study delves into post-COVID syndrome, particularly focusing on the prevalence of breathing pattern disorder (BPD) post-discharge and the subsequent recovery trajectory for most patients. The findings suggest a complex interplay of factors influencing dyspneic perception and breathing control in COVID-19 patients, urging further research to unravel these intricate mechanisms for improved clinical management and long-term outcomes.

Literature data link autonomic dysfunction in long COVID with persistent feelings of fatigue, chest oppression, depression, postural orthostatic tachycardia syndrome (POTS), and cognitive impairment following SARS-CoV-2 infection [25,26].

The Greek myth of Sisyphus provides a powerful representation of the experience of angor, the dysautonomic chest discomfort that can linger in long COVID patients. Sisyphus was condemned by the gods to an eternal punishment: he had to roll a massive rock up a hill, only to watch it roll back down when it reached the top, forcing him to start over in an endless cycle. Paraphrasing Sisyphus, one could say that “Sisyphus’ pandemic boulder” primarily affects the heart as well as the mind, and the challenge of modern cardiological pathophysiology should be the attempt to recognize the psycho-vegetative mechanisms of cardiac oppression that characterize the cardiac alterations present in PASC and in the distress of modern life.

Just as with Albert Camus’ Sisyphus, scientific research endeavors to transform detrimental endogenous and exogenous stimuli impacting cardiac contractility into a regenerative process of plastic modulation that restores ‘cardiac lightness’ and a state of well-being. Camus’ Sisyphus manages to transmute daily toil into a sense of happiness and adequacy: ‘I leave Sisyphus at the foot of the mountain!’ One always finds one’s burden again. But Sisyphus teaches the higher fidelity that negates the gods and raises rocks. He, too, concludes that all is well. This universe henceforth without a master seems to him neither sterile nor futile. Each atom of that stone, each mineral flake of that night-filled mountain, in itself forms a world. The struggle itself toward the heights is enough to fill a man’s heart. One must imagine Sisyphus happy.’” [27].

In the forthcoming sections, we will delve into the neural network governing cardiac activity by exploring the concept of cardiac neuromodulation. Our exploration will encompass the neurophysiological foundations of cardiac innervation, the principles underlying the modern ortho-parasympathetic integration concept, the emotional aspects of respiratory sinus arrhythmia (RSA), and the significance of identifying neurocardiac genes.

A comprehensive examination of both intrinsic and extrinsic cardiac innervation will undoubtedly enhance scientific research undertakings aimed at elucidating the neurovegetative mechanisms underlying dysautonomic disorders present in various diseases, including PASC syndromes.

## 2. From Aristotelian Cardiocentrism to the Cardiac Circulation Concept

The importance of the heart in human physiology was also emphasized in Aristotle’s (4th century BC) philosophical analyses [28]. He believed that only living beings possess a soul, which is inevitably reliant on the body. The soul (the form) cannot exist detached from the body (the matter) nor subsist independently. Aristotle attempted to unify the diverse psychophysical functions of the soul by attributing them to a single point of reference, the heart. He believed that all other bodily components were subservient to the heart and that the heart was the source of life (Aristotle, De juventute et senectute, 3, 469a–410), [29,30].

Galen (3rd century BC) rejected Aristotle’s cardiocentrism, but he reinforced the traditional notions that recognized the heart as the source of the body’s innate heat and the sole organ truly connected to the soul.

The concept of a mind–body connection in regulating the heart was first proposed by the Roman physician Aulus Cornelius Celsus in his work “De Medicina” around AD 30, where he noted that “fear and anger and any other state of mind may often be apt to excite the pulse” [31].

Cardiology advanced significantly in the 15th century, coinciding with the emergence of the first and most rigorous anatomical and physiological investigations. The pioneering research owes a considerable debt to the anatomical illustrations of Leonardo da Vinci and Vesalius. Leonardo’s diagrams proved more accurate in depicting the heart valves, while Vesalius erroneously posited the presence of pores in the interventricular septum.

In the 17th century, William Harvey [32] proposed the concept of cardiac circulation, which revolved around the movement of blood from the left to the right ventricle. He also scrutinized arteries, veins, and the heart’s septa [33]. Far more intricate accounts of circulation were provided by Servet [34] and Marcello Malpighi through his microscopic studies and the discovery of arterioles and capillaries [35].

## 3. The “Neurological” Heart

In his 1561 work *Observationes Anatomicae*, Gabriele Falloppia was the first to describe the existence of the cardiac nerve plexus beneath the aortic arch. However, he incorrectly connected the vagus nerve to the sympathetic trunk. Some scholars believe that his book *Observationes Anatomicae* is a collection of unillustrated annotations critical of Andreas Vesalius’s *De Humani Corporis Fabrica*. Vesalius attempted to refute Falloppia’s findings in a letter but was unsuccessful [36]. In response to one of Falloppia’s criticisms in 1564, Vesalius admitted to being aware of the ductus venosus and ductus arteriosus, even though he had not included them in his *Fabrica* [37]. Richard Lower (1631–1691), with his 1669 publication *De Corde*, was a leading figure in late 17th-century cardiology [38]. Lower began with a comprehensive description of the heart’s muscular anatomy and nervous system (Figure 4I) [39], and then proposed that blood flow was driven by heart contractions, asking, “If the blood moves through its own power, why does the heart need to be so fibrous and so well supplied with nerves?” [40]. Lower made another significant contribution to neuroscience by conducting an experiment with Thomas Willis on dogs to demonstrate that the heart’s control center is located in the hindbrain. They severed the nerves connecting the hindbrain to the heart, resulting in the heart being “quickly engorged with blood and died” [40]. This observation foreshadowed the discovery of cardiorespiratory nuclei in the brainstem. Lower experimentally verified that the heart is controlled by nerves by performing a ligation of the vagus nerve. The effect was linked to an increase in resting heart rate: “The heart, which before beat quickly and regularly, begins to palpitate and quiver as soon as the ligature is applied; the wretched animal prolongs a weary life for a day or two to the accompaniment of heart tremor and excessive dyspnoea, and finally dies without warning” [41]. Moreover, Lower demonstrated that small pieces of excised heart muscle continued to beat. It would take another 250 years for the myogenic theory of the heartbeat to be definitively proven. Albrecht von Haller, who succeeded Lower as professor in Göttingen in 1753, followed in his footsteps and took an anatomical interest in the heart and pericarditis. However, his most significant contribution was the discovery of the heart’s automatism and the excitation of muscle fibers.

Antonio Scarpa’s groundbreaking illustrations of the cardiac nerves in his 1794 work *Tabulae Nevrologicae* earned him a prominent position among the pioneers of modern neurology. His meticulous depictions and detailed anatomical observations challenged the prevailing theories of Albrecht von Haller and Johann Bernhard Jacob Behrends (1792) [42,43], who maintained that the heart’s rhythmic contractions were solely driven by its own intrinsic “irritability” and not influenced by neuronal activity. Scarpa’s work effectively refutes these erroneous notions.

While Scarpa did not specifically refer to ganglia in his text, his illustrations clearly revealed the presence of both ganglia and fusiform enlargements of the nerves, which he termed “corpora olivaria”, in the hearts of larger herbivorous mammals. These structures were not only observed at the base of the heart but also distributed across the surfaces of the ventricles (Figure 4II).

**Figure 4 biology-13-00266-f004:**
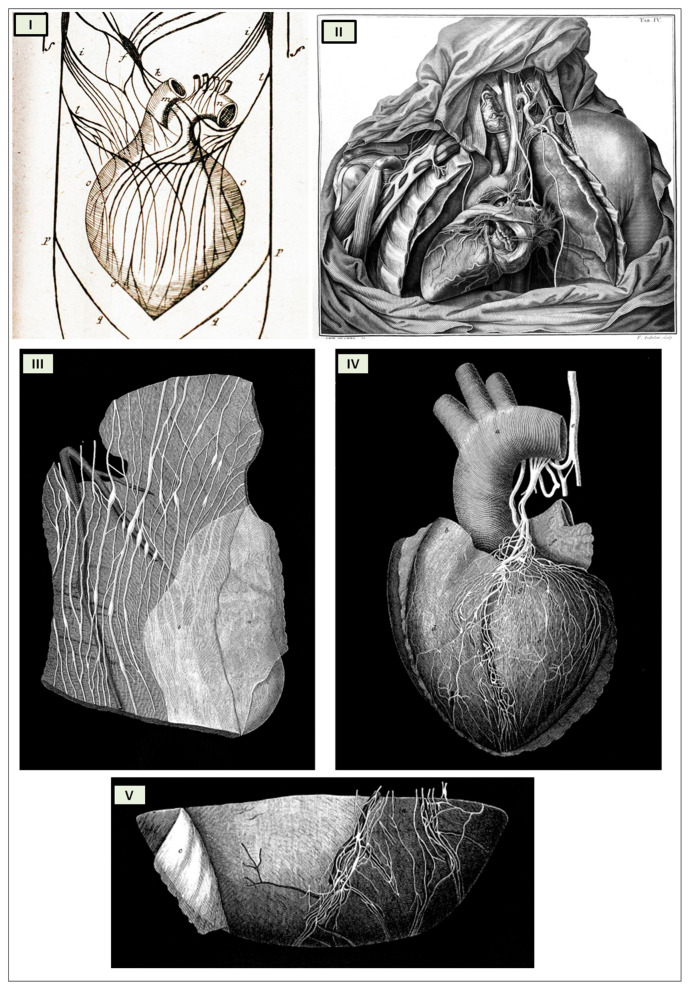
The ancient tabulae of the cardiac innervation. (**I**) Tabula A “f” Plexus Cardiacus [adapted from Richard Lower Cap I Cordis Anatome in Tractatus De Corde 1669, London [39]]; (**II**) Plate IV [adapted from Antonio Scarpa’s *Tabulae Nevrologicae* (https://www.nlm.nih.gov/exhibition/historicalanatomies/scarpa_home.html (accessed on 8 April 2022)), [44]; (**III**) Plate II, Figure 3 “The ganglia and nerves of the left ventricle of a heifer’s heart and cardiac fascia”; (**IV**) Plate I, Figure 1 “The nerves of the heart of a child nine years of age”; (**V**) Plate I, Figure 2 “The ganglia and nerves at the apex of the left ventricle of the sound human heart” [adapted from Lee, Robert. *On the Ganglia and Nerves of the Heart*. Philosophical Transactions of the Royal Society of London 139 (1849): 43–46. https://www.jstor.org/stable/108467 (accessed on 8 April 2022) [45]].

In 1849, Robert Lee, inspired by Scarpa’s anatomical insights, further refined the mapping and understanding of cardiac innervation. He conducted extensive dissections of human and bovine hearts at various ages and in conditions of myocardial hypertrophy, providing valuable insights into the distribution and function of cardiac nerves (Figure 4III–V). *“The drawings executed by Mr. West with the greatest pains and attention to accuracy, will supply the need of special verbal description of the nervous filaments, their anastomotic enlargements and fusiform swellings; and the series of my dissections show that the nerves of the heart which are distributed over its surface, and throughout its walls to the lining membrane and columnae carnese, enlarge with the natural growth of the heart, before birth and during childhood and youth, until the heart has attained its full size in the adult; that the nervous supply of the left ventricle is greater than that of the right; and that when the walls of the auricles and ventricles are affected with hypertrophy, the ganglia and nerves of the heart are enlarged like those of the gravid.”* [45].

Physiologists Theodor W. Engelmann (1843–1909) and Walter H. Gaskell (1847–1914) challenged the prevailing neurogenic theory of the heartbeat by conducting groundbreaking research that established the concept of myogenic genesis. Their meticulous experiments demonstrated that the isolated apex of a frog ventricle, devoid of ganglionic cells, could still generate rhythmic contractions when stretched [46]. This finding provided compelling evidence that the heart’s intrinsic properties, rather than external neural signals, were the primary drivers of its rhythmicity.

In 1923, Tullio Terni published his seminal work in the *Archivio Italiano di Anatomia e di Embriologia* (Italian Archive of Anatomy and Embriology), revealing the existence of a preganglionic nervous center in the thoracic-lumbar region of the spinal cord. This center, dubbed the “Terni column”, is a longitudinal column of nerve cells that spans from the first thoracic segment to the second lumbar segment [47]. Terni’s research demonstrated that this spinal center serves as a crucial source of sympathetic innervation for the cardiac system. The preganglionic neurons in the first five to six segments of the thoracic spinal cord, located in the lateral gray column, synapse with neurons in the superior cervical, medium cervical, and stellate ganglia, as well as neurons of the sympathetic trunk from the second to the fourth thoracic ganglia (Figure 5I).

Terni’s discovery of the spinal sympathetic innervation of the heart provided a critical piece of the puzzle in understanding the complex interplay between the nervous system and the cardiovascular system. His work not only challenged the traditional neurogenic theory but also paved the way for further research into the neural regulation of heart function.

Throughout fetal development, Terni’s column stimulates intermittent respiratory movements, thus enhancing the development of thoracic cavity compliance. This autonomic nucleus might also participate in the cardiovascular response to prenatal hypoxia, although its exact function remains unknown [48].

## 4. The Cardiac Neuromodulation

The autonomic nervous system stands out as a distinct anatomical and functional division of the nervous system. This distinction traces back to Marie François Xavier Bichat (1771–1802).

In his seminal work “*Physiological Researches upon Life and Death*” (1800), Bichat defined life as the sum of those functions that resist death and observed that living organisms constantly strive to maintain themselves in a state of equilibrium despite the destructive forces that surround them. Bichat further divided life into two distinct domains: organic life (“vie organique”), also known as the vegetative system, which corresponded to the process of nourishment, and animal life, (“vie animale”), or the animal system, which corresponded to the process of interaction with the environment.

Bichat conceptualized organic life as the essential functions of the heart, intestines, and other internal organs, controlled by a system of ganglia (“ganglions”), clusters of independent neural centers (little indipendent brains) in the chest cavity. In contrast, animal life pertained to symmetrical organs (eyes, ears, and limbs), governed by habits and memories but primarily driven by intelligence and astuteness. This aligned with the brain’s characteristic attributes, yet it would not be feasible without the heart, the cornerstone of organic life. Bichat’s use of the term “animale” harkens back to the Latin noun “anima” or soul [52]. In this manner, Bichat effectively shifts the traditional paleochristian conception of the soul’s abode from the “heart” to the “brain”, a hallmark of the Enlightenment era.

Building upon Bichat’s foundational work in delineating the autonomic nervous system, 19th-century researchers made significant strides in elucidating the specific neural mechanisms governing the heart’s function and cardiac autonomic control.

In 1846, the Weber brothers, Eduard Weber and Ernst Heinrich Weber, published their findings on the inhibitory effects of vagus nerve stimulation on the heart rate in their work “Muskelbewegung” [53,54,55].

Concurrently, in the 1850s, researchers such as Carl Stelling [56], Claude Bernard [57], and Charles-Édouard Brown-Séquard [58] identified the sympathetic nerves as pressor (blood pressure-raising) nerves, in contrast to the vagus nerve’s inhibitory effects, as documented in their works. Building on this, in 1866, Carl Ludwig [59] and then Ewald Hering [60,61], described the vasomotor reflexes, which involve the autonomic regulation of blood vessel diameter and blood pressure.

In 1886, Langley employed nicotine and other medications to meticulously investigate the sympathetic and parasympathetic nervous systems, which he subsequently unified under the concept of the ‘autonomic nervous system’ [62].

Finally, in 1946, Ulf Svante von Euler demonstrated that the sympathetic transmitter substance was noradrenaline (norepinephrine), a significant discovery in understanding autonomic neurotransmission in the cardiovascular system, as published in his paper “A Specific Sympathomimetic Ergone in Adrenergic Nerve Fibres (Sympathin) and its Relations to Adrenaline and Noradrenaline” [63].

These pioneering researchers laid the foundation for the modern understanding of the autonomic nervous system’s influence on cardiac function and homeostasis.

Several regions of the neuraxis orchestrate the central control of orthosympathetic and parasympathetic output. This intricate network of central autonomic connections plays a pivotal role in regulating visceral functions and maintaining homeostasis by adapting to internal and external stimuli. The central autonomic system is hierarchically organized into four distinct levels: spinal, bulbopontine, pontomesencephalic, and forebrain (Figure 6 and Figure 7) [64,65,66,67,68,69].

The cardiac preganglionic parasympathetic axons of the vagus nerve (Figure 6 in red) primarily arise from medullar neurons placed in the dorsal motor nucleus of the vagus (DMV), expanding presumably wider into the ventrolateral portion of the nucleus ambiguus (NAmb). It has been suggested that preganglionic parasympathetic axons originating from neuronal somata adjacent to NAmb are B fibers slowing rapidly the heart rate, conduction, and force of myocardial contraction. A second group of cardiac parasympatethic preganglionic axons originates from the dorsal motor nucleus (DMV), and these are the C fibers propagating slower neural impulses. Selective stimulation of the DMV axons affects atrioventricular conduction and ventricular contraction and reduces the heart rate (HR); however, it starts off more slowly and exhibits a more varied pharmacology than what is observed after stimulating the vagus nerve’s B fibers. The two groups of neurons have somewhat different discharge patterns, apart from conduction speed. While the DMV neurons show a non-uniform, non-respiratory-dependent discharge and are unresponsive to baroreceptors and chemoreceptors, the cardiac neurons next to NAmb exhibit a respiratory rhythm and receive input from both sources. These features of medullar parasympathetic neurons have implied the theory that they presumably reflect the differential functions of the two groups. An experiment that targets distinct populations of intrinsic cardiac ganglionic cells by nerve terminals from the proper DMV and those near the NAmb, each anterogradely labeled with a different fluorescent tracer, supports this theory [70]. It is also possible that DMV neurons innervate small, intensely fluorescent (SIF) cells in the ganglia. On the contrary, Jones [71] has hypothesized that the tonic input transmitted by the more slowly moving C fibers may interconnect with the rhythmic respiratory input carried by the faster vagal B fibers on the same cardiac ganglia.

At the spinal cord level, segmental sympathetic or sacral parasympathetic reflexes mediate specific responses to stimuli, albeit under the influence of higher brain regions. These reflexes play a crucial role in maintaining cardiovascular, respiratory, gastrointestinal, and micturition functions. The lower brainstem, also known as the bulbopontine level, plays a pivotal role in reflex control of core physiological processes. It regulates circulation, respiration, gastrointestinal functions, and micturition through a network of interconnected neurons. The upper brainstem, referred to as the pontomesencephalic level, engages in sophisticated pain modulation and behavioral integration in response to stress. It orchestrates physiological and psychological adaptations to challenging situations. The forebrain level encompasses the hypothalamus, the master regulator of the autonomic nervous system. It interacts with the anterior limbic circuit, a network comprising the insula, anterior cingulate cortex, and amygdala, to modulate emotional and visceral responses.

The cortex of the insula functions as the primary interoceptive cortex and combines sensations of the viscera, pain, and temperature. The dorsal insula has an organization that is specialized for visceral processing and acquires inputs from gustative, visceral, muscle, and skin receptors through the thalamic region. Through their connections with limbic and neocortical association areas, the neurons of the dorsal portion of the insula extend to the right anterior insula, which connects these interoceptive signals with emotional and cognitive elaboration. As a result of this integration, one can consciously sense their body. Additionally, the insula acts as a visceromotor area and controls both sympathetic and parasympathetic outputs, primarily through a relay in the lateral hypothalamus.

The anterior cingulated cortex is connected to the anterior portion of the insula and is divided into ventral (emotional and default mode network), and dorsal (cognitive and frontoparietal awareness networks) regions. Subcallosal and precallosal regions make up the cortical ventral anterior part of the cingulate gyrus, which is connected to the brainstem, prefrontal cortex, amygdala, hypothalamus, and insula on an extensive level. Through their connections, the anterior cingulate cortex regulates autonomic systems.

The amygdala enriches the neuronal sensory afferents with an affective and/or emotional connotation through a downstream neuronal network with which it participates in the neuroendocrine and autonomic responses to stress. Through its extensive connections with the hypothalamus and brainstem, including the periaqueductal gray and the medullary reticular region, the central nucleus of the amygdala (CeA) plays a key role in the coordination of stress responses, particularly alarm responses. The hypothalamus is a center for visceromotor activity and starts selective patterns of autonomic and endocrine responses based on various stimuli such as variations in the temperature of the blood, hypoglycemia, osmolarity, or outside agents of stress. The preoptic-hypothalamic area is divided into three functional regions: periventricular, medial, and lateral. The periventricular zone comprises the circadian pacemaker, localized in the suprachiasmatic nucleus, and numerous regions implicated in neuroendocrine management through the pituitary gland. The medial zone includes the medial preoptic area, dorsomedial nucleus (DMH), and paraventricular nucleus (PVN), which are involved in osmoregulation, thermoregulation, and stress reactions. The PVN, DMH, and lateral hypothalamic areas are the primary autonomic efferences of the hypothalamus. The PVN contains several groups of neurons that are selectively activated during stress responses, including magnocellular neurons that release arginine-vasopressin (AVP) into the systemic circulation, neurons that release corticotropin-releasing hormone and stimulate the adrenocortical axis, and neurons that project to the brainstem and spinal cord autonomic nuclei. The PVN regulates stress reactions, food and sodium intake, glucose metabolism, and cardiovascular, renal, gastrointestinal, and respiratory functions through these efferences. The DMH is involved in stress responses, thermoregulation, and cardiovascular control. Arousal, eating, and reward-driven behaviors are all regulated by hypocretin/orexin neurons in the posterior lateral hypothalamus.

The periaqueductal gray matter of the midbrain (PAG), the parabrachial nucleus (PBN) of the pons, and various medullary regions, such as the nucleus of the solitary tract (NTS), the ventrolateral reticular formation of the medulla, and the medullary raphe (Figure 6 and Figure 7) are all involved in autonomic output [65]. The PAG is composed of several longitudinal columns that coordinate the micturition reflex, participate in cardiovascular responses related to respiratory regulation, and modulate pain through their diverse spinal, brainstem, and cortical connections. The PAG integrates forebrain and lower brainstem activity in response to tasks including stress, pain control, and somatic adaptative changing. The PBN is an important location for relaying information to the hypothalamus, amygdala, and thalamus from the spinal cord’s converging visceral, thermoreceptive, and discomfort stimuli. The PBN in particular plays a significant role in the regulation of the digestive, cardiac, and breathing systems. The NTS has multiple subnuclei that are arranged “viscerotropically” (i.e., with a special affinity for various organs) and acts as the first relay station for flavor and visceral afferent information. Therefore, taste inputs are allowed in the rostral region of the NTS, gastrointestinal afferents are allowed in the intermediate part, and baroreceptor, cardiac, chemoreceptor, and pulmonary afferents are allowed in the caudal part. The NTS sends this neuronal input to the upper brainstem and forebrain areas, either directly or via the PBN. As a result, the NTS serves as the first central “switch” station for all medullary reflexes controlling the function of the heart (baroreflex and cardiac reflexes), the breathing process (carotid chemoreflex and pulmonary mechanoreflexes), and digestive tract peristalsis. The rostral ventrolateral medulla (RVLM), which contains the C1 group of adrenergic neurons holding epinephrine, is a crucial region for the control of arterial blood pressure. The sympathoexcitatory RVLM neurons collect and combine a wide range of inputs from the brainstem and forebrain, particularly the hypothalamus, including the PVN. The efferent neurotransmission of glutamate in the RVLM sends direct and rhythmic stimulation to sympathetic preganglionic neurons, which govern cardiac function and peripheral resistance in general. The cardiopulmonary reflexes, baroreflexes, and chemoreflexes belong to the reflexes under the control of the RVLM. These reflexes are mediated by inhibitory GABA neurons in the distal ventrolateral medulla, which receives inhibitory signals from the baroreceptor-sensitive neural pathway in the NTS.

GABAergic neural populations of the caudal ventrolateral medulla preserve a rhythmic, constantly slow inhibitory manage on the upper ventrolateral medulla and retransmit the inhibitory signals from the NTS, producing the inhibitory element of the sympathetic arterial baroreflex. Data from neural stimulations show that the caudal medulla holds pressure-dependent regions too. Furthermore, the A1 neuronal population in the distal ventrolateral medulla releases norepinephrine to the hypothalamus and is implicated in a reflex route that causes the production of vasopressin or arginine vasopressin (AVP) in reaction to hypovolemia or hypotension.

The upper portion of the ventromedial medulla, which includes the caudal raphe nuclei, is important for body temperature control, pain regulation, and autonomic respiratory control. Through input to preganglionic sympathetic neurons, which cause effective cutaneous vasoconstriction and thermogenesis without shivering in brown adipose tissue, one group of medullary raphe neurons initiates adrenergic responses to cold.

The maintenance of circulation, temperature control, and displacing of localized blood flow during stress and exercise depend heavily on the sympathetic efferences. Pregangliar neurons, which are mostly found in the intermediolateral cell column of the thoracolumbar spinal cord’s T1 to L2 segments, are the source of sympathetic efferences. The intermediolateral neurons are organized into segregated functional units that give the innervations to specific subgroups of sympathetic ganglion neurons and collect different separate portions of afferent inputs, triggering segmental somatic and viscerosympathetic reflexes. Premotor neurons in the brainstem and hypothalamus recruit different pregangliar sympathetic units in an integrated manner to start particular patterns of responses to particular internal or external events that cause stress, such as exposure to extreme temperature, low blood sugar, dehydration, changes of posture, physical activity, and/or stress. The principal sources of premotor sympathetic innervation include the RVLM, medullary raphe, A5 noradrenergic neuronal population of the pons, PVN, and lateral hypothalamic region.

Opposite to the sympathetic system, which affects multiple effectors, the parasympathetic system mediates organ-specific reflexes. Most vagal preganglionic parasympathetic neurons, which are arranged viscerotropically and innervate local ganglia in the liver, pancreas, enteric nervous system (ENS), and respiratory tract, are found in the DMV. The DMV generates all vago-vagal responses controlling digestive tract movement and secretion after receiving information from the NTS. The dorsal motor nucleus is the source of a substantial population of cardiac vagus nerves (Figure 6).

The NTS activates the cardiovagal neurons next to the NAmb during the baroreflex and inhibits them during inspiration, as will be covered in greater detail later. For life in a constantly changing environment and to maintain homeostasis, interactions between the heart and the brain are crucial (Figure 8) [72,73]. Neurons located in the lateral gray matter at the S2–S4 segments of the sacral spinal cord are the source of the sacral preganglionic output. With synchronized interactions with both lumbar sympathetic neurons at the T12-L2 levels and somatic motor neurons of the Onuf nucleus at the S2–S4 levels, which innervate the external urinary sphincter and pelvic floor, these neurons play a crucial role in normal defecation, micturition, and sexual organ function (Figure 6) [64].

Cardiac function is controlled neurally by the anterior insular region, anterior part of the cingulate gyrus cortex (ACC), amygdala, hypothalamus, periaqueductal gray matter, parabrachial nucleus, and several medulla areas. Further, these regions are connected to homeostatic reflexes, responses to emotions, and stress reactions. By means of the sympathetic and parasympathetic neural systems, they govern and regulate heart rate (HR) and cardiac contractility (Figure 9) [75]. It is still unclear if this central control might be lateralized.

The heart’s intrinsic electrophysiological properties arise from pacemaker activity generated by specialized cardiomyocytes within the cardiac intrinsic conduction system, comprising the sinoatrial (SA) node, atrioventricular (AV) node, bundle of His, and Purkinje fiber network. The HR, excitability, and contractile function of these cardiomyocytes depend on the interplay between their intrinsic characteristics and regulation by the vagus and sympathetic nerves via the intrinsic cardiac ganglionated plexus, or intrinsic cardiac nervous system (ICNS) [76].

The HR (chronotropism) is regulated by the SA node’s spontaneous depolarization (automatism), which is controlled by a “voltage regulator”. The voltage regulator is produced by the cyclic activation and deactivation of different membrane ion channels, and a “calcium regulator” that is triggered by the rhythmic release of Ca^2+^ from the sarcoplasmic reticulum via the ryanodine receptor 2 (RYR2). The rhythmic increase in cytosolic Ca^2+^ activates the Ca^2+^-Na+ exchanger current, leading to depolarization [77,78,79].

The cardiac cycle starts with depolarization spreading through connexin channels to neighboring cardiomyocytes, followed by the opening of voltage-gated Na+ (Nav1.5) channels. These channels are rapidly inactivated by depolarization, which activates both L-type calcium channels accountable for the action potential plateau and voltage-gated K+ channels responsible for repolarization. The synchronized activity of these channels produces the excitability of the His-Purkinje system (bathmotropism), the velocity of AV conduction, or dromotropism (PR interval), and the duration of the cardiac action potential (QT interval). Systolic contraction (inotropism) occurs through excitation-contraction coupling, when calcium released from the sarcoplasmic reticulum through RYR2 binds to the troponin complex and activates the contractile machinery. Cellular relaxation during diastole (lusitropism) occurs when cytosolic Ca^2+^ is removed by the sarcoendoplasmic reticulum Ca^2+^-ATPase (SERCA) uptake pump, which is negatively regulated by the protein phospholamban (an inhibitor of Ca^2+^-ATPase).

The cardiac nervous system is coupled to areas distributed throughout the neuraxis (Figure 8 and Figure 9) and includes intrinsic and extrinsic neural networks. Morphologically, the ICNS is an intrinsic neural plexus with epicardial ganglia, as described by Lee in 1849 (Figure 4III–V). Its function is controlled by extrinsic influences mediated by the vagal and sympathetic nerves. A variety of different kinds of neurons are present in the intrinsic cardiac ganglia, most likely including the following subtypes: afferent neurons, motor (parasympathetic and sympathetic) neurons, and linking local circuit neurons (interneurons), which make up the first three types of neurons. Vagal or sympathetic inputs regulate their intrinsic reactivity. The modulation of cardio-cardiac reflexes is significantly influenced by this network of ganglionic cells (Figure 10 and Figure 11) [80].

When the sympathetic efferent route is activated, the parasympathetic pathway is often inhibited, and vice versa, according to the obsolete theory of an autonomic balance. The dorsal motor nucleus of the brainstem and the spinal cord, respectively, both possess preganglionic efferents of the parasympathetic and sympathetic systems that innervate the heart. Sympathetic transmission occurs in the sympathetic ganglion chain near the spinal cord, from preganglionic (dashed) to postganglionic (solid). Preganglionic parasympathetic axons synapse with postganglionic parasympathetic components in the cardiac plexus, which is reached by postganglionic axons from the sympathetic trunk.

Furthermore, it is evident that certain physiological reactions can entail the simultaneous cardiac stimulation of both sympathetic and parasympathetic activation. In comparison to vagal regulation, the development of adrenergic innervation of the heart occurred relatively later in the course of evolution [it is absent in elasmobranch fish]; hence, the simultaneous activation of both autonomic pathways may be very important in expanding the heart’s functional capacity to fulfill the metabolic requirements of the body within constantly changing behavioral and environmental circumstances.

Using immunohistochemical methods, it has been evidentially proven that a range of neurochemical substances are present. Epicardial ganglia, as well as intracardiac ganglia situated within the boundaries of the heart hilum, exhibit immunoreactivity (IR) towards neurotransmitters and neuromodulators such as choline acetyltransferase (ChAT), the enzyme accountable for the synthesis of acetylcholine (ACh); tyrosine hydroxylase (TH), the enzyme which catalyze the production of the sympathetic neurotransmitter noradrenaline (NA); vasoactive intestinal peptide (VIP, recognized to co-release alongside ACh); neuropeptide Y (NPY, reputed to co-liberate together with NA); neuronal nitric oxide synthase (nNOS, involved in NO synthesis by parasympathetic, sympathetic, and non-adrenergic non-cholinergic (NANC) nerves) [83] (Figure 12); synaptophysin (a tracer for presynaptic fibers); substance P (sub P); and calcitonin gene-related peptide (CGRP).

Most ganglion cells utilize acetylcholine as their primary transmitter.

SIF cells are TH-positive and are often found inside larger ganglia, in tiny clusters, or scattered along the atrial and ventricular walls. The exact function of SIF cells in the heart is still unclear, since the majority of them are not adjacent to axons and presumably are not under neural control, according to numerous observations by Pauza and his co-workers [87,88,89,90]; Pauziene et al., [86,91].

An intriguing feature of the cardiac ganglia is the presence of biphenotypic neurons, i.e., neurons showing both ChAT-IR and TH-IR. These neurons account for 10 to 20% of all intrinsic cardiac neurons. Cholinergic cardiac neurons may also exhibit immunoreactivity for nitric oxide, i.e., such cholinergic neurons should be considered biphenotypic nitrergic [92,93] This feature of producing and perhaps releasing both neurotransmitters is most interesting and needs further investigation [82,94].

As mentioned above, with respect to the extrinsic sympathetic innervation, the cardiac preganglionic axons originate from neurons of the intermediolateral (IML) nucleus of the Terni’s column of the spinal cord (Figure 5II). Rhythmic excitatory glutamatergic inputs from neurons of the rostral ventrolateral medulla (RVLM) are collected by IML neurons. The cardiac preganglionic sympathetic neurons are neurons that liberate acetylcholine and project to neurons with norepinephrine release of the superior and middle cervical, cervicothoracic (stellate), and first four thoracic ganglia, which give rise to axons that innervate the heart via the superior, middle, and inferior cardiac nerves [49]. The right-to-left map of sympathetic nerves is asymmetrical [51] and shows differences in expression between individuals; this may clarify their heterogeneous effect on cardiac electrophysiologic properties.

The cardiovagal innervation represents the extrinsic parasympathetic output. The preganglionic cardiovagal neurons are cholinergic, and their axons extend to the cardiac ganglia through superior cervical, inferior cervical, and thoracic rami, which merge with adrenergic nerves of the heart to form the extrinsic cardiac plexus in the mediastinum. Vagal nerve fibers distribute within the walls of the atria and ventricles, in the sinoatrial (SA) and atrioventricular (AV) nodes [50].

Sympathetic activation elicits an increase in inotropism and dromotropism, faster conduction through the AV node, a boost in the stimulation of the His-Purkinje system, which determines a strengthening of contraction force during systole, and faster relaxation of the cardiac muscle cells during diastole.

The primary impact of the vagus, on the other hand, is suppression of the pacemaker activity of the SA node (reduction in heart rate, HR), decreased AV conduction, and diminished excitability of the His-Purkinje system by cholinergic neurons of the cardiac ganglia.

The tonic vagal control of the SA node automatism prevails over that of the sympathetic system during resting conditions. HR has a circadian pattern; it increases in the early morning because the sympathetic activity surges and then decreases during sleep, particularly during non-REM (rapid eye movement) sleep, for vagal predominance. However, phasic transient vagal interruption and sympathetic activation result in HR surges during REM sleep. Vagal activity rapidly decreases in response to orthostatic stress, hypovolemia, or exercise. In conditions with very low basal HR (e.g., athletes, during non-REM sleep, or patients with sinusal bradycardia), vagal activation could unexpectedly increase HR by reducing the interval between depolarizations of the atria. The vagal activation, particularly in the ventricles, is higher in the setting of prominent simultaneous adrenergic activation; this so-called “accentuated antagonism” depends on presynaptic inhibition of sympathetic transmission.

HR variability (HRV) results from interactions between the vagal and sympathetic influences on the SA node. HRV can be evaluated during deep breathing and during the Valsalva maneuver.

The cardiac preganglionic sympathetic IML neurons are tonically activated by premotor glutamatergic sympathoexcitatory neurons of RVLM, which act as a common effector of descending and reflex pathways controlling blood pressure (BP) and cardiac function; some of these neurons also synthesize epinephrine (C1 group). Psychological stress, pain, hypoxia, hypovolemia, and hypoglycemia activate RVLM neurons both directly and via descending inputs from the forebrain. RVLM is inhibited by the baroreflex via disynaptic inhibition from the NTS, mediated by GABA (gamma-aminobutyric acid) neurons of the caudal portion of the ventrolateral medulla (Figure 8 and Figure 5II). The NAmb contains the majority of cardioinhibitory vagal motoneurons that control SA automatism and AV node conduction. These neurons are activated by glutamatergic inputs from barosensitive neurons of the NTS and inhibited by local GABAergic neurons and by GABAergic neurons of the medullary ventral respiratory group that are active during inspiration. In this way, the vagal control of the HR is modulated on a beat-to-beat basis by respiration; cardiovagal activity is lower during inspiration and higher during expiration. This physiologic condition is known as respiratory sinus arrhythmia (RSA). RSA is a phenomenon that occurs during both regular breathing and enhanced breathing patterns like sighs [95]. It is considered a healthy aspect of HRV and is thought to enhance the efficiency of gas exchange [96] or help reduce cardiac workload while maintaining optimal blood gas levels [97]. Sighs and the accompanying RSA typically occur at the beginning of arousal and are believed to play a role in reopening the airways after obstruction [98]. In young, healthy individuals, RSA is often evaluated as a high-frequency HRV, reflecting the combined activity of the sympathetic and parasympathetic nervous systems. Even in full-term infants, cardiorespiratory coupling, as seen in RSA, is present from the early stages of life. This coupling strengthens with gestational age, indicating a shift towards parasympathetic dominance after birth. Conversely, premature infants may exhibit immature cardiorespiratory coupling, characterized by lower HRV in the high-frequency range, which can impair responses to stress and increase the risk of sudden death [95]. Therefore, RSA is an important measure of cardiovagal output and health and declines linearly with age. In addition, the Hering–Breuer reflex generated by pulmonary mechanoreceptors via the NTS may contribute to the RSA. The caudal region of the solitary nucleus receives afferents mostly from vagal and glossopharyngeal afferents from baroreceptors, cardiac receptors, chemoreceptors, and pulmonary receptors. Respectively, the vagal afferents by aortic baroreceptor and chemoreceptor, cardiac and visceral sensory receptors in most organs of the thoracic and abdominal cavities with cell bodies in the nodose ganglion (NG), together with carotid baroreceptor and chemoreceptor of glossopharyngeal afferents with cell bodies in the petrosal ganglion (PG), provide inputs to the nucleus of the solitary tract. This nucleus begins a variety of cardiovascular reflexes and also carries cardiovascular receptor inputs to the thalamus and parabrachial nucleus. Therefore, it is the primary essential station for all reflexes of the medulla, inclusive of the baroreflex and cardiac reflexes controlling BP and HR.

The baroreceptor reflex (baroreflex) is a fundamental BP buffering tool and is activated by the mechanical deformation of vessel walls in the carotid sinus and aortic arch during systole. An increase in BP stimulates baroreceptor afferents of the glossopharyngeal and vagus nerves, which activate the NTS via a monosynaptic excitatory input. The barosensitive NTS neurons initiate sympathoinhibitory and cardioinhibitory responses with two different pathways. The sympathoinhibitory pathway controls total peripheral resistance via disynaptic inhibition of RVLM neurons mediated by gamma-aminobutyric acid neurons of the caudal ventrolateral medulla. Through the second pathway, the cardioinhibitory signal elicits a decrease in HR via direct excitatory inputs from the NTS to cardiovagal neurons of the NAmb. Numerous cardiovascular reflexes are triggered by afferents from the heart, coronary, and pulmonary arteries. Among these are cardiac unmyelinated afferents to the thoracic dorsal root ganglia, which run along the adrenergic nerves and provide input to the dorsal horn (primarily lamina I) and intermediate gray matter of the spinal cord, as well as myelinated and unmyelinated vagal afferents with cell bodies in the nodose ganglion, which provide input to the NTS. On the other side, myelinated vagal afferents are activated by atrial distension due to an increase in blood volume, which triggers the reflex of sympathetic input to the SA node, resulting in an increase in HR, as well as suppression of renal adrenergic activity and arginine vasopressin production, favoring salt and water excretion. Furthermore, unmyelinated spinal and vagal afferents innervating the ventricles are stimulated by strong mechanical or chemical stimuli, in particular products of ischemia or inflammation such as adenosine triphosphate, serotonin, and prostanoids.

Spinothalamic projections from lamina I neurons elicit the sensation of cardiac pain; these afferents can also trigger excitatory cardiac reflexes via local interneurons projecting to the IML (the so-called “cardio-cardiac reflex”). In response to the chemical stimulation of myocardial injury, unmyelinated vagal afferents in the ventricles may trigger a decrease in BP and HR (Bezold–Jarisch reflex). Stimulation of pulmonary arterial baroreceptors at physiologic pressure produces reflex vasoconstriction and respiratory stimulation. This could be implicated in cardiovascular control during exercise or in hypoxic conditions.

The cortex of the insula (IC), ACC, amygdala central nucleus (CeA), and numerous nuclei of the hypothalamus (Figure 8) project to the medullary and spinal nuclei to control the activity of the heart. These projections are either direct or indirect via a relay in the periaqueductal gray. Afferent cardiovascular information carried by the dorsal horn (layer I) or NTS neurons reaches cortical areas through the thalamus. Visceral afferents are also conveyed to the parabrachial nucleus of the pons, which relays the information to the thalamus, hypothalamus, and amygdala, and to catecholaminergic neurons of the A1/C1 group of the ventrolateral medulla (Figure 6). The role of these areas and the possible hemispheric lateralization of the control of cardiac function are yet poorly understood. The IC has been implicated in a wide range of functional activities as well as the underlying causes of several neurologic diseases. It is separated into two zones: dorsocaudal and rostroventral. The dorsocaudal zone contains many areas that receive thalamic-relay gustatory, viscerosensory, somatosensory, pain, and vestibular afferents. The rostroventral zone, which is connected to the ACC and the amygdala, is primarily engaged in affective neural elaboration. Electrical stimulation of the insula in patients undergoing surgical treatment for intractable epilepsy elicits a variety of visceromotor phenomena, including changes in BP and HR. Oversimplifying, stimulation of the left IC more frequently elicited a small decrease in HR and BP, whereas stimulation of the right IC elicited the opposite effect. These findings, further supported by fMRI studies, suggest that the left IC primarily regulates the parasympathetic and the right IC the sympathetic influence on the heart (Figure 5III) [50].

The “laterality hypothesis” has a “cardiotopic” counterpart of asymmetric autonomic nerve distribution around the heart. This complex rotation around the heart axis may be correlated to the remodeling/regression of the right aortic arch during embryonic development, whereas the left-sided arch persists and/or to the cardiac tube rotation during the formation of the four embryonic heart chambers. Typically, the right vagus regulates the activity of the cardiac atria, whereas the left vagus controls the activity of the ventricles. HR is therefore regulated by asymmetric sympathetic and parasympathetic innervation of the sinoatrial (SA) node. In factual terms, the right stellate cardiac nerve is the primary source of sympathetic input from the sinus atrial node, whereas the right vagus is the primary source of parasympathetic SA input. Contrarily, left side inputs are primarily responsible for controlling conduction time in atrioventricular fibers, which is enhanced by parasympathetic activity and reduced through sympathetic activity, again in both systems. However, left-side sympathetic stimulation is primarily responsible for increasing contraction of the myocardium. In conclusion, the right side primarily regulates cardiac frequency, whereas the left side mostly controls ventricular activity and pulse pressure. (Figure 5IV). The evaluation of asymmetric anatomical and functional autonomic nerve distribution in humans is important to understand anomalies of the heart with left-right predominance and to identify new treatments [50,99,100].

The ACC integrates autonomic responses with behavioral arousal via its wide associations with the IC, amygdala, prefrontal cortex, hypothalamus, and brainstem autonomic nuclei. Functional MRI (Magnetic Resonance Imaging) studies show that the ventral ACC is involved, as described above, in the “default mode network”, activated in the resting state in conditions of self-monitoring. Whereas the dorsal ACC, together with the anterior IC, is a central element of the so-called “salience network”; it is primarily recruited during tasks that demand cognitive control, such as conflict resolution, and is connected with an enhancement in sympathetic conduction, resulting in an HR increase. On the contrary, the subgenual ACC and adjacent ventromedial prefrontal cortex are inactivated at the same time. Pharmacological and neuroimaging studies show that subgenual ACC stimulation depends on HRV mediated by the vagus nerve, particularly in the right hemisphere.

The amygdala provides emotive meaning to inputs from sensory systems and is engaged in fear-conditioning processes. The amygdala includes several nuclei, among them the basolateral nuclear complex and the CeA. The medial aspect of the CeA projects to the hypothalamus and brainstem and triggers the autonomic, endocrine, and motor manifestations of fear responses. Sympathoexcitatory responses involve excitatory connections to the RVLM and inhibition of barosensitive neurons of the NTS. Functional neuroimaging studies have demonstrated a coactivation of the lateral and medial amygdala in relation to changes in HRV both at rest and during emotional tasks. The orbitofrontal and ventromedial prefrontal cortices offer an inhibition of the amygdala via the GABAergic neuronal population in the lateral CeA and in the intercalate nucleus between the basolateral amygdala and the CeA; these prefrontal modulations are involved in mechanisms of emotional regulation, including fear extinction. In fact, in addition to promoting vagal output, these prefrontal areas tonically inhibit sympathoexcitatory responses initiated in the amygdala.

Sympathoexcitatory responses during stress are mediated by projections from the hypothalamus. It tunes autonomic output to the heart via inputs that originate primarily from the PVN, DMH, and lateral hypothalamic area; these hypothalamic projections reach the periaqueductal gray parabrachial nucleus, RVLM, NAmb, DMV, NTS, and IML. THe hypothalamus modulates the baroreflex via NTS or NAmb connections. Experimental studies indicate that the inputs from the IC to the hypothalamus are mostly ipsilateral.

Lower HRV and baroreflex sensitivity (BRS) are important markers of cardiovascular risk, including an increased incidence of ventricular arrhythmias in individuals with primary cardiac disease, due to the fact that they reflect deficiencies in forebrain vagal or sympathetic drive, brainstem reflexes, or vagal or sympathetic output (as occurs in diabetic or amyloid neuropathy). Reduced HRV, mainly at the expense of decreased high-frequency HF (vagal) component and sometimes associated with indices of increased sympathetic activity, has been described in patients with ischemic stroke, epilepsy, multiple sclerosis, and Parkinson’s disease. An alteration in cardiac autonomic control, manifested by a reduction in HRV, has been correlated with sudden infant death syndrome (SIDS) and sudden unexpected death in epilepsy (SUDEP). Furthermore, new evidence suggests that a similar reduction in HRV is associated with long COVID [101]. The observed decrease in HRV in patients experiencing persistent long COVID symptoms highlights dysautonomia. This underscores the critical need for in-depth research to investigate interventions, both pharmacological and non-pharmacological, aimed at addressing this dysautonomia [102].

Central autonomic abnormalities can cause a wide range of cardiac arrhythmias, some of which are life-threatening. Vagal hyperactivity, for example, causes bradyarrhythmias such as AV block, whereas sympathetic overdrive causes both supraventricular and ventricular tachycardia, and both sympathetic and vagal hyperstimulation can cause atrial fibrillation (AF). Sympathetic activity in the ventricles is proarrhythmic, whereas vagal activation is antiarrhythmic.

Seizures and cardiac dysfunction are characterized by a number of shared features, including: cardiac arrhythmias as an ictal event that can localize the seizure focus, seizures and cardiac arrhythmias as co-occurring manifestations of channelopathies, and cardiovascular dysregulation and ictal arrhythmias as a potential mechanism underlying sudden unexpected death in epilepsy (SUDEP). These features suggest that seizures and cardiac dysfunction may be linked to common underlying mechanisms. Sinus tachycardia occurs in 80% to 100% of patients before, during, or after a temporal lobe seizure; paroxysmal AF, supraventricular or ventricular tachycardia, and ventricular fibrillation may also occur. Furthermore, temporal-lobe seizures may lead to ictal bradycardia and asystole. Seizures, more common in patients with left-sided seizures, can also produce alterations of cardiac repolarization as manifested by acute changes in the corrected QT (QTc) interval, which may be in part due to ictal hypoxemia. Mutations in genes encoding Na+ or K+ channels are associated with the coexistence of long QT syndrome (LQTS) or Brugada syndrome and epilepsy. Autonomic cardiovascular manifestations of seizures may have a role in the pathophysiology of sudden unexpected death in epilepsy, but this causal relationship is yet to be established.

Autonomic nervous system dysfunction is a widespread occurrence that relates significant heart diseases to neurologic problems. Many neurologic diseases, such as ischemic stroke, sudden cardiac death, and epilepsy, may have higher morbidity due to these severe autonomic effects on the heart.

## 5. Light-Heartedly

At the end of our review, we wish to propose a point of convergence between the sacredness of psychostasis, its social and artistic expression, and the science of cardiac innervation. We believe that they converge in the biological nature of contemporary neuroscience and introduce us to the neural mechanisms of cardioprotection and resilience.

As we discussed previously, a prevailing notion posits the concept of autonomic balance, whereby the activation of sympathetic and parasympathetic efferents is mutually inhibitory. While this holds true for the baroreceptor reflex, there is evidence to suggest that certain physiological responses may elicit simultaneous activation of both systems to safeguard the heart. In fact, the regulation of heart function is often governed by the synergistic interplay of sympathetic and parasympathetic stimulation, mirroring the combined control of urine production, penile erection, and the activity of secretory glands by both autonomic branches. For instance, combined stimulation of the sympathetic and vagal fibers in the heart can result in bradycardia (by peripheral chemoreceptors), tachycardia (by somatic nociceptors), or biphasic rhythms in the cardiac frequency (during the startle response). The cardiac response elicited during a dive provides a prime example of the potential physiological implications of combined sympathetic and parasympathetic activation. Provoked bradycardia is a remarkable physiological adaptation that markedly lowers cardiac oxygen consumption, effectively shielding the heart from the detrimental effects of hypoxia. Given that bradycardia significantly diminishes ventricular contractions due to non-neural mechanisms, it is plausible that augmenting the influx of noradrenergic signals to the ventricular myocardium could counterbalance the frequency-dependent contractility reduction, thereby optimizing stroke volume and emulating the conditions induced by the chemoreflex. Another method might be increased sympathetic outflow via the stimulation of beta-adrenoreceptors in coronary arterioles, which may reduce the resistance of the coronary artery in order to sustain blood flow and oxygen release to the heart. A hypothesized explanation for this combined effect involves the likely release of neuropeptides in the ICNS or the activation of SIF cells in the ganglionic population of the heart. SIF cells have extensive vagal innervation in the cardiac ganglia, and their fluorescent activity is confined to vesicles carrying catecholaminergic neurotransmitters. Therefore, the discharge of catecholamines from these vesicles may be the cause of the paradoxical vagally mediated tachycardia [70,103,104,105,106] (Figure 13). One of our authors remains skeptical about the theory of vagal innervation of SIF cells, as this is in sharp contrast to what is seen in analogue cells of the adrenal medullas. Although in rare cases nitrergic axons are adjacent to SIF cells, Pauza has not identified parasympathetic (cholinergic) nerve fibers in close vicinity to cardiac SIF cells across many animals, such as rats, mice, guinea-pigs, rabbits, pigs, sheep, and humans [107] (Figure 14). Schematic representations of the two hypotheses are provided in Figure 13 and Figure 14.

Yet another plausible explanation for heart-activity-dependent tachycardia is the release of a co-transmitter from the vagus nerve, accompanied by a distinct pattern of co-vagal neuropeptide transmission under various conditions. Moreover, it can be posited that dysregulation of this cardioprotective balance system can act as both a cause and an outcome of numerous cardiac pathologies [103].

Resilience has bestowed an additional evolutionary advantage upon the psyche-soma relationship, entailing an individual’s ability to swiftly bounce back and adapt to adversity and/or stressful conditions while safeguarding psychophysiological assets. Greater resilience is correlated with faster cardiovascular recuperation following subjective emotional experiences, manifested as reduced perceived stress, enhanced recovery from illness or trauma, and improved management of chronic pain and early dementia. Compromised resilience stems from autonomic nervous system dysregulation, which can be assessed by measuring vagal regulation via RSA. As we’ve already mentioned, RSA is the rhythmic variation in heart rate that is linked to breathing and is brought on by the vagal nerve’s activity. The heart rate increases during inhalation when vagal activity is reduced, but at some point during expiration, vagal activity is reintegrated, causing the heart rate to slow down. The activation of a branch of vagal fibers that originates from the medullary NAmb and terminates at the sino-atrial node of the heart is the primary mechanism by which RSA is induced.

As proposed by Stephen Porges’s 1995 polyvagal theory (PVT), the NAmb vagal fibers represent novel evolutionary adaptations that have emerged to facilitate complex emotional responses and social interactions among animals. Accordingly, RSA measures are considered to offer a non-invasive window into how this relatively recent NAmb vagal system engages with emotional experiences [108,109].

PVT postulates the existence of three distinct neural networks within the sympathetic and parasympathetic nervous systems that respond to the perception of risk (i.e., safety, danger, threat to life) in a hierarchical manner governed by an evolutionary principle. This order of activation aligns with the Jacksonian principle of dissolution, which suggests that newer adaptation networks are initially engaged, and if they prove insufficient, older systems are recruited [110]. From this perspective, the three autonomic systems form a phylogenetically ordered hierarchy of responses [111].

Consequently, the three polyvagal neural networks are associated with social communicative behavior, the defensive strategy of “aggressive” mobilization, and defensive immobilization. Specifically, they represent: 1. The ventral vagal complex (VVC) underpins the brain mechanisms that mediate the “social engagement system”, attitudes toward others, and relationships. The NAmb, the motor component of the VVC, governs and controls the muscles of the pharynx, larynx, and possibly the bronchi and heart. 2. The fight-or-flight response, the fundamental and earliest defense mechanism employed by animals, is primarily associated with the sympathetic nervous system. This defensive strategy necessitates heightened metabolic output to facilitate locomotor behavior. When the VVC fails to quell the perception of danger, the sympathetic system assumes hierarchical precedence. Upon activation, the sympathetic nervous system elicits a cascade of physiological changes, including increased muscle tone, redistribution of blood to major muscle groups, suppression of gastrointestinal function, dilation of bronchioles, augmented heart rate and respiration, and the release of catecholamines. 3. Emerging from the DMV, the dorsal vagal complex (DVC) delivers the primary vagal motor fibers to the subdiaphragmatic organs and innervates the cardiac ganglia. This circuit, designed to respond adaptively to an overwhelming threat or fear, represents the most primitive (i.e., evolutionarily earliest) stress response. When activated, the DVC triggers a passive reaction characterized by a decrease in muscle tone, a significant reduction in cardiac output to conserve metabolic resources, and changes in intestinal and bladder function via reflexes leading to defecation and urine excretion. This visceral suppression represents an effort to minimize metabolic and oxygen demands to the bare minimum for survival, particularly in its cardioprotective role. This response often manifests as behavioral shutdown, collapse, and a “freezing” response in humans and can be interpreted as a dissociative state that can lead to loss of consciousness.

Therefore, saying that a resilient person faces life lightly has a physiological value because this person shows a strong aptitude to activate the new parasympathetic system (VVC) and stimulate left hemispheric predominance. (Figure 15) [111].

## 6. Discussion

The paper delves into the historical evolution of beliefs surrounding cardiac activity, tracing back to ancient Egyptian concepts of postmortem judgment and the weighing of good and bad deeds. It further explores how these ideas transcended cultures, influencing Greek, Islamic, and Christian beliefs in the form of psychostasis. The discussion extends to modern interpretations of a “light heart” and the physiological implications associated with emotional states like fatigue and depression.

This comprehensive review has meticulously traced the historical trajectory of medical and neuroanatomical research on cardiac innervation, shedding light on the nascent biomolecular concepts of cardiac neuromodulation. This endeavor underscores the unremitting progress of anatomical science, poised to be further enriched and corroborated by genetics and genomics studies in the foreseeable future. The recent identification of neurocardiac genes has emerged as a paradigm-shifting discovery. These genes typically encode ion channel proteins, simultaneously expressed in both the autonomic nervous system and the myocardium. Alterations in these genes now appear to be implicated in diverse instances of sudden cardiac and neurological demise [113]. As such, this line of inquiry warrants vigorous pursuit. The neurocardiac axis stands as the cornerstone of the nascent scientific understanding of psychostasis. Recently, the polyvagal theory has been leveraged to illuminate the significance of the neurocardiac axis and its interplay with the COVID-19 pandemic [114,115] and other respiratory infections, culminating in autonomic nervous system dysregulation.

Additionally, the article discusses the autonomic dysfunction in COVID-19 patients, including symptoms such as fatigue, chest oppression, depression, POTS, and cognitive impairment following SARS-CoV-2 infection. The Greek myth of Sisyphus is used as an analogy for the long-term experience of angor, the dysautonomic chest discomfort that can persist in long COVID patients. The authors stress the importance of scientific research in transforming harmful stimuli into a regenerative process that restores “lightness of heart” and overall well-being, similar to the philosophy of Albert Camus’s Sisyphus.

In summation, further research is imperative to unravel the intricate implications of the neurocardiac axis for psychostasis, long COVID, and long non-COVID-19 ARIs.

## 7. Conclusions

In conclusion, the interdisciplinary analysis presented in this paper bridges ancient beliefs with contemporary scientific understanding of cardiac neuromodulation. By examining the historical roots of psychostasis and its influence on various cultures, alongside modern interpretations of cardiac function and emotional well-being, this study sheds light on the intricate interplay between cultural beliefs, physiological responses, and neurological mechanisms. This holistic approach not only enriches our understanding of cardiac innervation but also underscores the enduring significance of ancient concepts in shaping contemporary perspectives on heart health and well-being.

## Figures and Tables

**Figure 5 biology-13-00266-f005:**
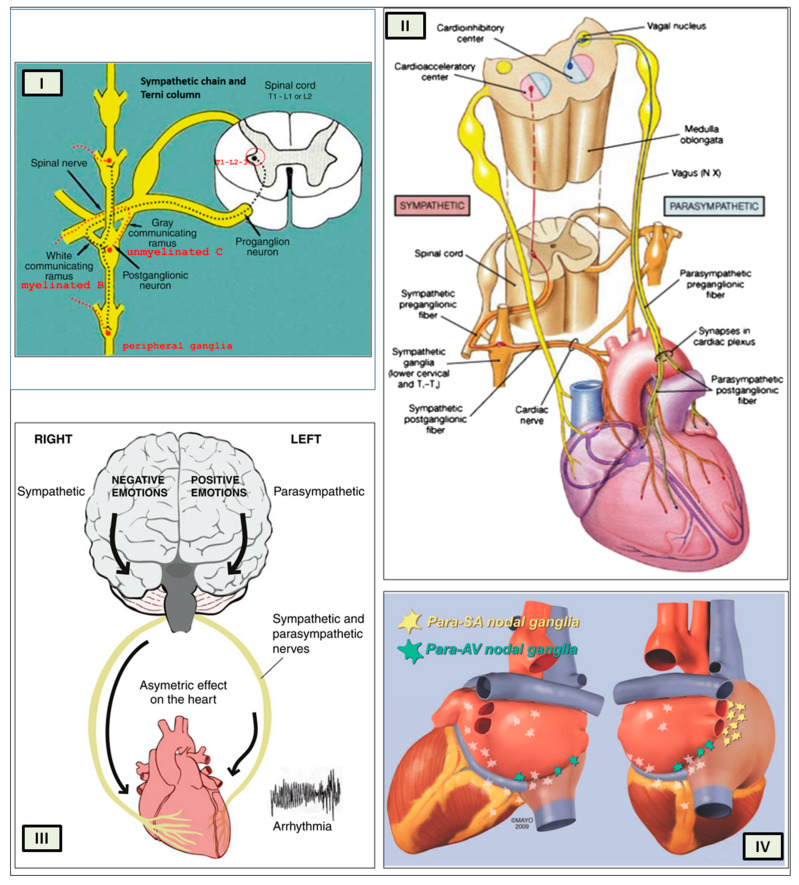
The modern “tabulae” of the cardiac innervation. (**I**) Preganglionic nervous center in the thoracic-lumbar region of the spinal cord, situated in intermediolateral nucleus in humans [adapted from Thiene et al., 2019 [48]]; (**II**) Anatomic representation of the extrinsic-cardiac nerve fibers of the autonomic nervous system—the cardio-acceleratory sympathetic and cardio-inhibitory parasympathetic systems [adapted from Martini, Nath 2006, in Olshansky et al., 2008 [49]]; (**III**) The “laterality hypothesis”. According to one concept of cortical representation of certain emotions, negative emotions (such as anger and fear) are processed largely in the right hemisphere, whereas good emotions (such as happiness) are processed primarily in the left hemisphere. The right hemisphere is predominantly related to sympathetic activity, whereas the left hemisphere is connected to parasympathetic activity. Most nerve traffic traveling from the brain to the heart is ipsilateral [adapted from Taggart 2013 [50]]; (**IV**) “laterality hypothesis” and asymmetrical autonomic cardiac innervation. While the left vagus mostly governs ventricular function, the right vagus primarily regulates atrial function. The right-side stellate cardiac nerve is the primary source of sympathetic SA input, respectively, and the right vagus is the primary source of parasympathetic SA input. Therefore, ventricular regulation and pulse pressure are primarily left-sided functions, while autonomic control of heart rate is primarily a right-sided function. [adapted from Kapa 2010 [51]].

**Figure 6 biology-13-00266-f006:**
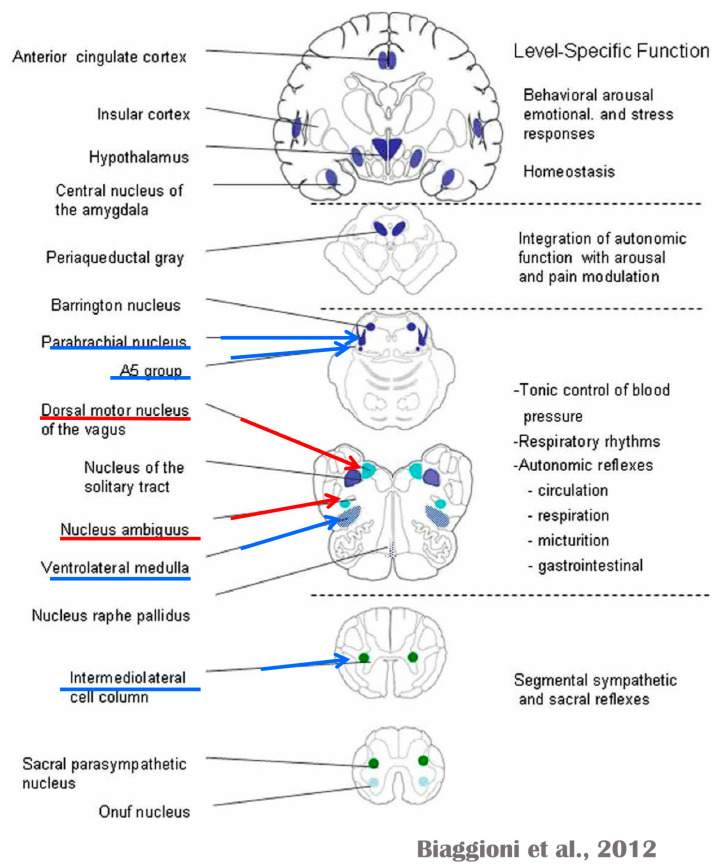
Central autonomic control areas and levels of interaction of autonomic control [adapted from Benarroch 2012, chapter 2 from Biaggioni’s book *Primer on the Autonomic Nervous System* [64]]. The central control of the sympathetic and parasympathetic nervous systems involves interconnected areas throughout the neuraxis. The central autonomic network plays a crucial role in regulating visceral functions, maintaining homeostasis, and adapting to internal and external challenges. This network operates at four hierarchical levels: spinal, bulbopontine, pontomesencephalic, and forebrain. The spinal level controls segmental reflexes, the bulbopontine level regulates circulation, respiration, and other functions, the pontomesencephalic level integrates autonomic control with pain modulation and stress responses, and the forebrain level includes the hypothalamus for integrated autonomic and endocrine responses. Additionally, components of the anterior limbic circuit in the forebrain are involved in integrating bodily sensations with emotional and goal-related autonomic responses. Sympathetic fibers are highlighted in blue, while parasympathetic fibers are highlighted in red.

**Figure 7 biology-13-00266-f007:**
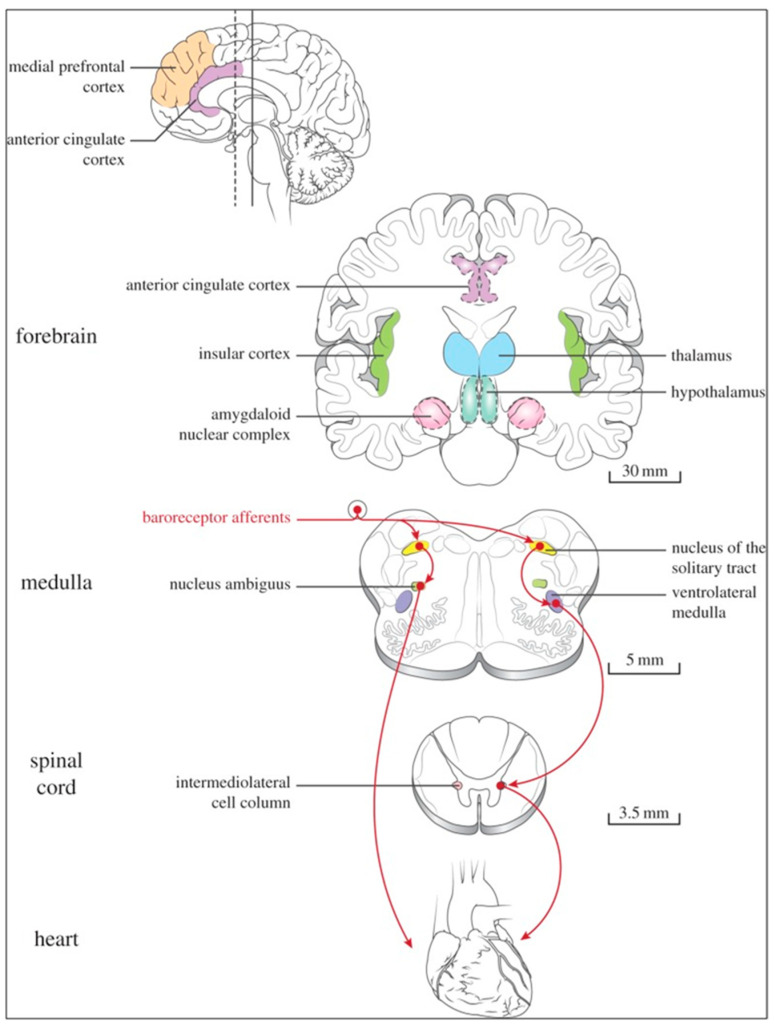
Crucial areas of brain–heart interactions. A number of regions of the forebrain, brainstem, and spinal cord related to autonomic function in humans are shown. The baroreceptor reflex, which mediates the homeostatic control of blood pressure, is illustrated by arrows; the parasympathetic and sympathetic outflows are indicated by the left and right branches of the baroreceptor afferents, respectively. Scale bars show the human brain’s approximate dimensions. [adapted from Chang et al., 2016 [65]].

**Figure 8 biology-13-00266-f008:**
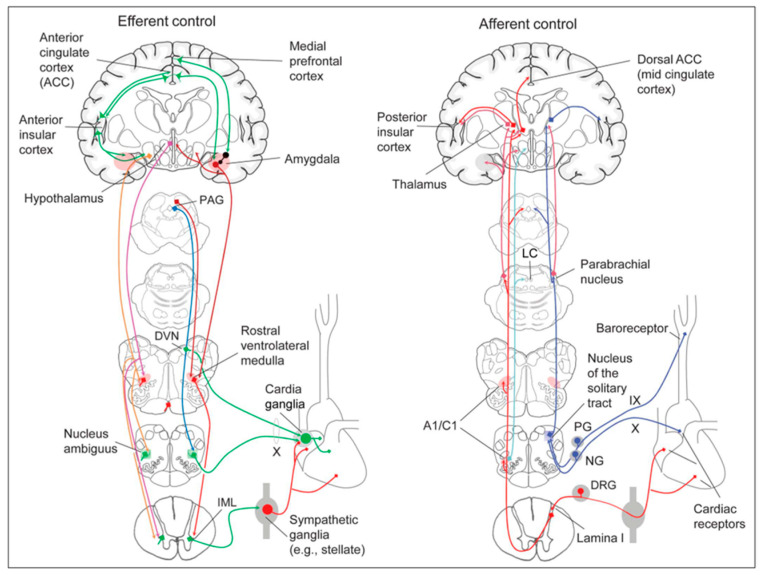
Neural control of the heart is integrated at all levels of the neuraxis [adapted from Palma and Benarroch 2014 [74]].

**Figure 9 biology-13-00266-f009:**
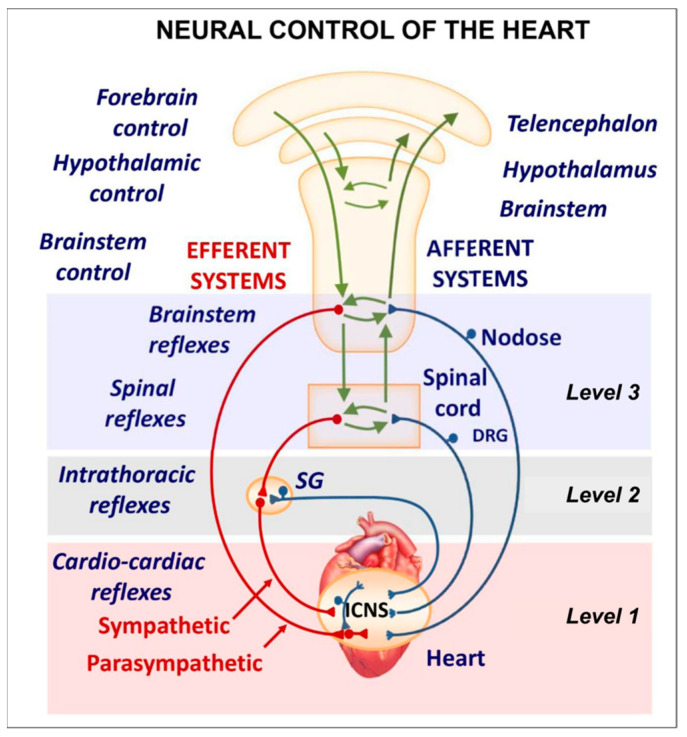
Heart activity regulation by the brain. The central nervous system (level 3), intrathoracic extra-cardiac neurons (levels 2 and 3), and intracardiac neurons (level 1) all play a role in the autonomic regulation of the heart. [Adapted from Hanna et al., 2017 [75]] SG stands for sympathetic ganglion, DRG for dorsal root ganglion, and ICNS for intracardiac nervous system.

**Figure 10 biology-13-00266-f010:**
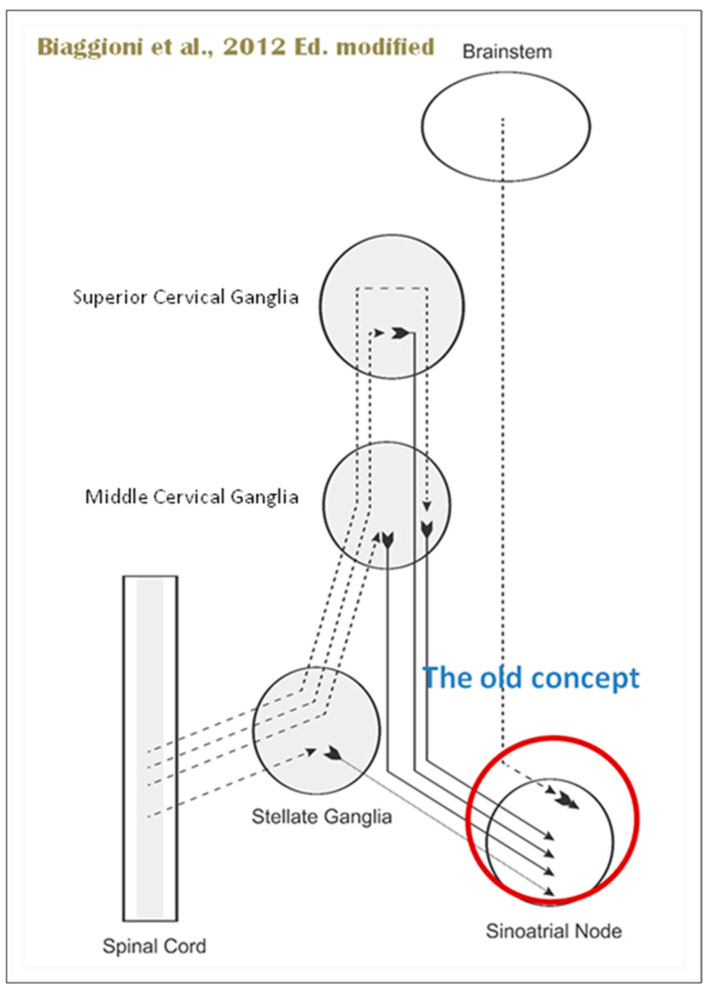
The old concept of autonomic cardiac control [adapted from Javier G. Castillo, David H. Adams (2012) Cardiac Vagal Ganglia, chapter 37 from Biaggioni’s Book *Primer on the Autonomic Nervous System* [81]]. The heart receives input from preganglionic sympathetic and parasympathetic nerves originating from the spinal cord and brainstem’s dorsal motor nucleus, respectively. Sympathetic nerve transmission involves a transfer from preganglionic (dashed) to postganglionic (solid) neurons at the sympathetic ganglion chain near the spinal cord. Postganglionic sympathetic neurons then travel to the cardiac plexus, where they interact with preganglionic parasympathetic neurons. Autonomic nerves originating from the cardiac plexus have the ability to form synapses with intrinsic cardiac neurons situated in the network of cardiac ganglia.

**Figure 11 biology-13-00266-f011:**
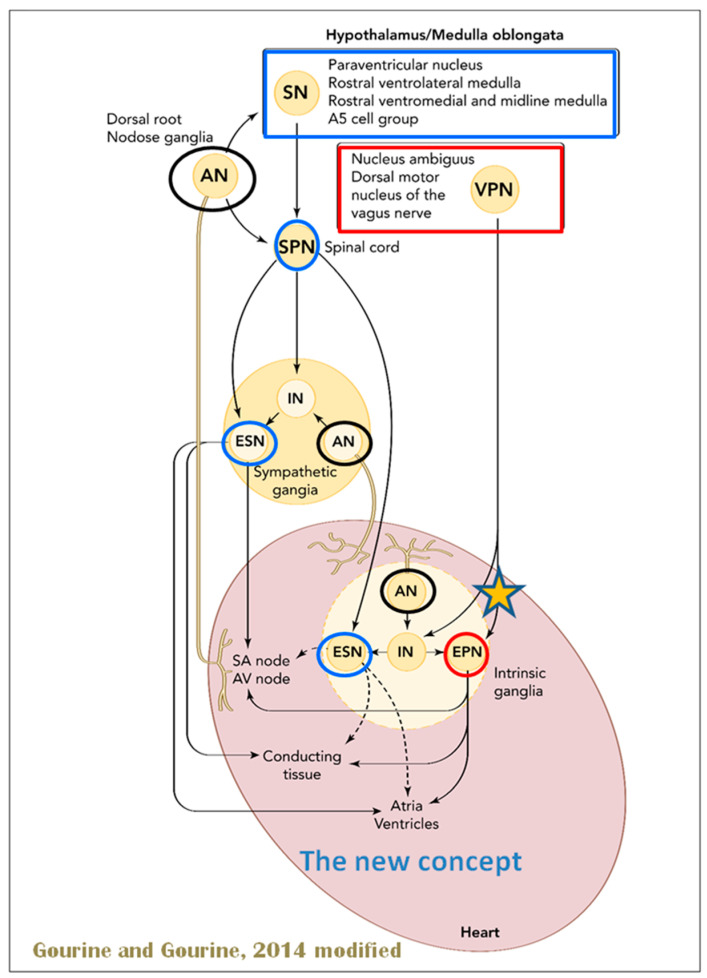
The hypothetical new concept of autonomic cardiac control [adapted from Gourine and Gourine 2014 [82]] The heart’s sensory and efferent neural circuits are represented. **AN** stands for **afferent** (sensory) neurons; **ESN** stands for **efferent sympathetic** neurons; **EPN** stands for **efferent parasympathetic** neurons; **VPN** stands for **vagal** (parasympathetic) **preganglionic** neurons; **IN** stands for **interneurons**; **SN** stands for **sympathoexcitatory** neurons, and **SPN** stands for **sympathetic preganglionic** neurons.

**Figure 12 biology-13-00266-f012:**
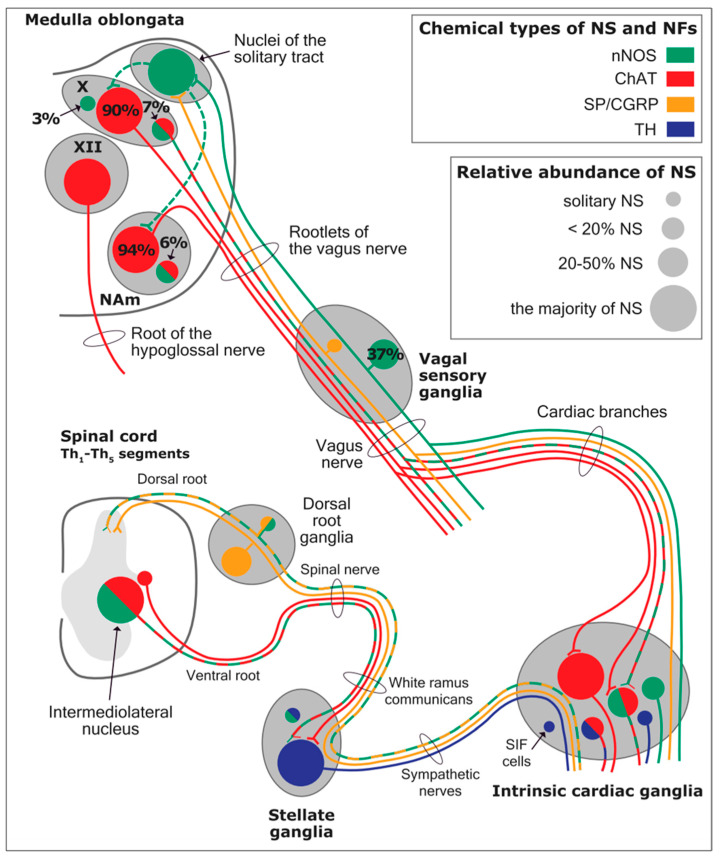
The distribution of different chemical phenotypes of neuronal somata (NS) and nerve fibers (NFs) among the presumed sources of extrinsic cardiac innervation [modified from Navickaite et al., 2021 [83]]. The nodose ganglion contains a significant population of nitrergic neuronal somata (37%); this ganglion is most likely the principal source of nitrergic nerve fibers throughout the vagal nerve and its cardiac branches. However, approximately 7% of neuronal cell bodies in the dorsal vagal nucleus co-express both ChAT and nNOS, indicating that this nucleus may be the origin of a small number of cardiac nitrergic nerve fibers. Furthermore, the dorsal vagal nucleus (DNV) contains tiny interneurons that are positive for nNOS (about 3%), physically resemble conventional interneurons, and most likely act around the nucleus. In the nucleus ambiguous (NAm), approximately 6% of neuronal somata display the biphenotypic characteristics of nNOS and ChAT, although only cholinergic nerve fibers have been observed exiting the nucleus. In both the dorsal vagal nucleus and nucleus ambiguous, a network of nNOS-IR nerve fibers is identified, which, according to existing literature, may originate from the solitary tract nuclei (indicated by the dashed line), where a crowded group of little nitrergic neuronal somata is found. Most sympathetic neuronal somata in the intermediolateral nucleus of the spinal cord co-express choline acetyltransferase and neuronal nitric oxide synthase, indicating that neuronal nitric oxide synthase is involved in the neuromodulation of preganglionic sympathetic neurotransmission. Nevertheless, only a single neuronal soma in the stellate ganglia has positivity for both tyrosine hydroxylase and neuronal nitric oxide synthase. Nitrergic neuronal somata are likewise rare in the dorsal root ganglia, suggesting that merely solitary nitrergic nerve fibers for the heart may come from this ganglionic station. Studies by Rysevaite et al. (2011) [84] and Pauziene et al. (2015, 2017) [85,86] form the basis for the anatomical map of the intrinsic cardiac ganglia’s makeup. Respectively, the following abbreviations (DNV), stand for the dorsal nucleus of the vagal nerve, (X), the hypoglossal nucleus, (Nam), the nucleus ambiguous, and (SIF cells), the tiny, highly fluorescent cells [83].

**Figure 13 biology-13-00266-f013:**
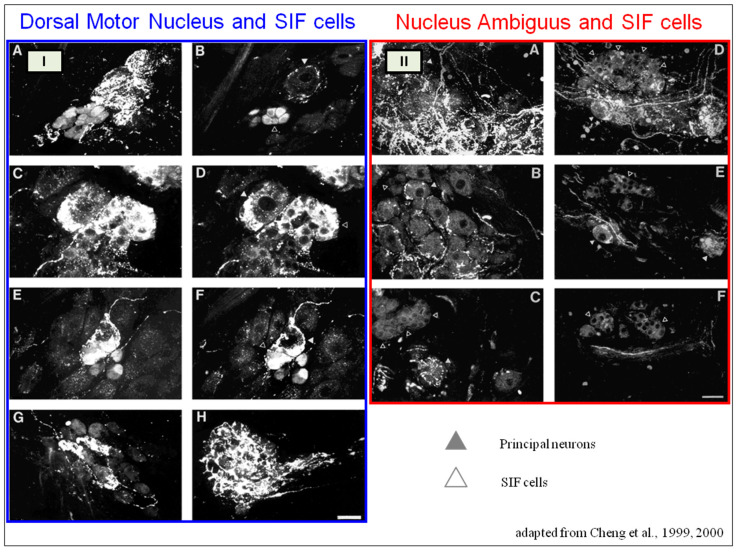
Vagal nerve and cardiac SIF cells. (**I**) Dorsal motor nucleus of the vagus (DMV) endings on small, intensely fluorescent (SIF) cells. Confocal images illustrating how some DMV motor fibers innervate SIF cells (open arrowhead), some innervate principal neurons (PNs, solid arrowhead), and some innervate both [adapted from Cheng et al., 1999 [104]]; (**II**) Axon terminals from the nucleus ambiguus (NA) did not project to SIF cells. According to a confocal image stack, PNs have extremely dense terminals, whereas SIF cells do not [modified from Cheng et al., 2000 [70]; this may indicate that the NA and DMV motor neuron pools have different functional specializations. I: panel (**A**) shows a composite projection of confocal optical sections that illustrates a case where PNs in the upper right pole are heavily innervated by DiI-labeled varicosities from the DMV, while the adjacent SIF cells in the lower left pole are not innervated by these DMV fibers. Panel (**B**) shows an optical section from the stack of sections projected in panel A. This section clearly illustrates the different morphology between the PNs and the SIF cells. Specifically, the image shows a PN that is encircled with some DiI-labeled varicosities, indicated by the solid arrowhead. In contrast, a cluster of SIF cells is identified with the open arrowhead, demonstrating their distinct morphology compared to the PNs. Panel (**C**) depicts a composite projection of optical sections that illustrates a single DMV axon providing innervation, via varicosities, to both a PN and a SIF cells. Panel (**D**): The image shows a PN with DiI-labeled varicosities and a cluster of SIF cells that are innervated by tagged DMV varicosities. Panels (**E**,**F**) show a composite projection and optical section of a DMV axon that innervates a PN and a nearby cluster of SIF cells. Panel (**G**,**H**) are composite projections of confocal images that show DMV axonal innervation in isolated SIF cell clusters without PNs. In II: Based on the detailed search results provided, the key points regarding the relationship between the NA axon endings and the SIF cells are: The NA axon endings do not project to or innervate the SIF cells in the cardiac ganglia. This is clearly demonstrated across multiple panels in the image description: panel (**A**) shows a projection of confocal images with very dense NA axon endings on the PNs, but not on the adjacent SIF cells. Panels (**B**,**C**) further delineate the close contacts of the DiI-labeled NA axon endings on the PNs, while the SIF cell clusters remain uninnervated. Panel (**D**) provides another example showing that the NA motor fibers innervate only the PNs and not the SIF cells, which are clearly visible but lack any innervation. Panels (**E**,**F**) confirm this finding, with the solid triangles indicating the innervated PNs, while the SIF cells (open triangles) remain devoid of NA axon endings. In summary, the search results demonstrate that the NA axon terminals specifically target and innervate the PNs in the cardiac ganglia, but do not project to or form synaptic connections with the SIF cells.

**Figure 14 biology-13-00266-f014:**
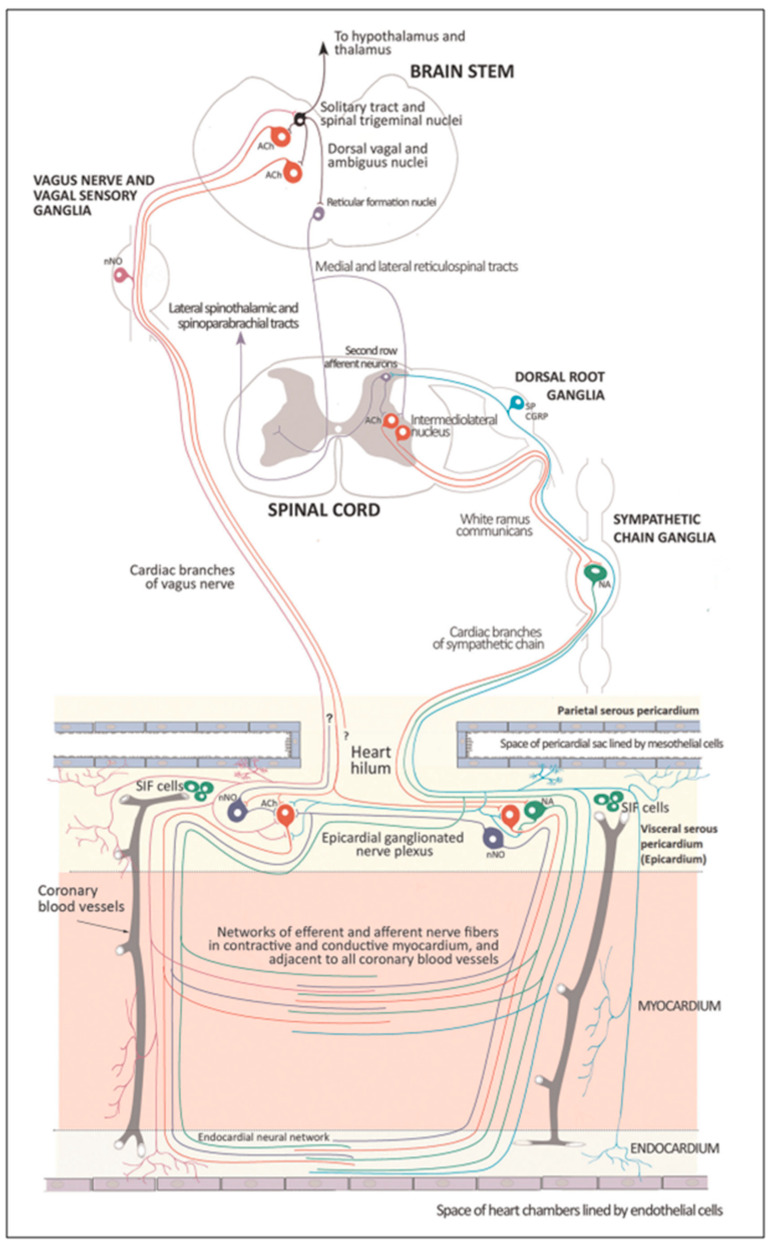
A representation of the mammalian autonomic nervous system of the heart in schematic form. The illustration demonstrates how preganglionic axons of the sympathetic system leave the spinal cord, connect with second-order sympathetic neurons in the sympathetic chain or, most likely, within the intrinsic cardiac ganglia, and then continue as postganglionic adrenergic axons to supply signals to the various heart cells (depicted in green). Second-order parasympathetic neurons in the epicardial ganglionated nerve plexus (shown in brown) form synapses with the preganglionic axons of the vagus nerve (in brown). While the first four thoracic dorsal root ganglia give origin to axons containing substance P (SP) and calcitonin gene-related peptide (CGRP), cardiac sensory neurons located in the dorsal root and vagus sensory ganglia mostly have nitrergic axons spread to the heart. The cardiac hilum serves as the conduit for all nerves traveling to the heart. Several nitrergic neuronal somata may be found in the epicardial ganglionated nerve plexus, with neuronal nitric oxide (nNO) connecting the neurons in the brain stem and spinal trigeminal nuclei (indicated in blue). Consider seeing the ganglia-adjacent tiny, intensely fluorescent (SIF) cells that synthesize noradrenaline (NA) and adrenaline. Despite the ganglionated nerve plexus’s epicardial placement, the heart’s muscle tissue and endocardium have an incredibly dense interconnected system of sensory and efferent nerve fibers [adapted from Aksu et al., 2021 [107]].

**Figure 15 biology-13-00266-f015:**
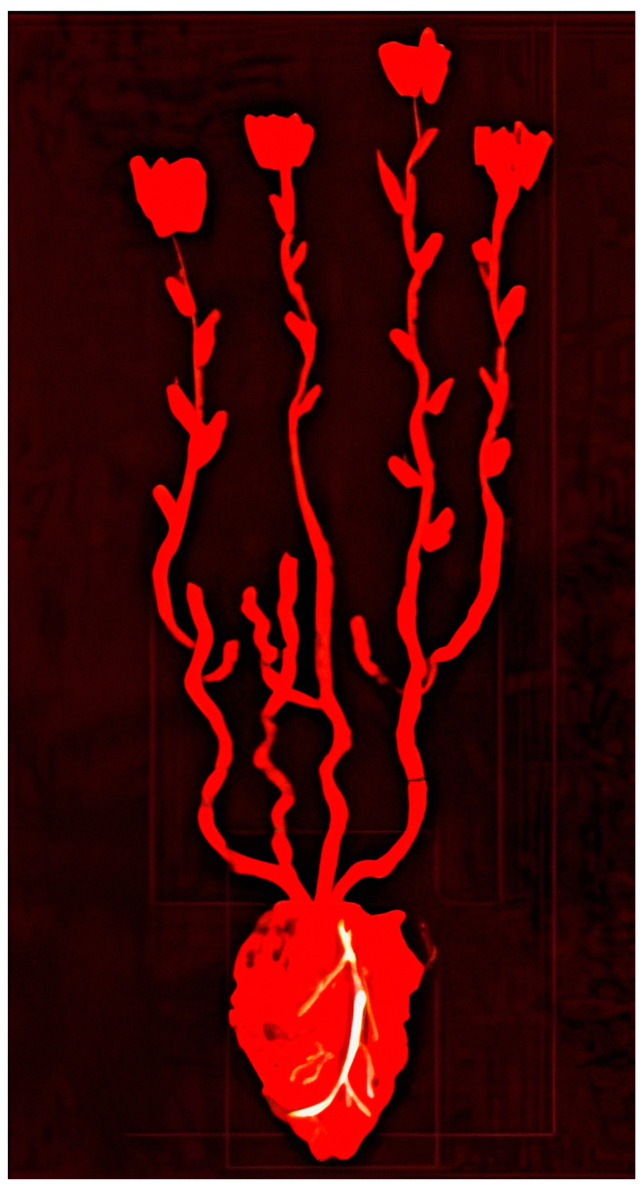
“Light-Heartedly” is loosely based on the illustration by Massimo Dezzani [adapted from Givone 2022 [112]].

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
