# Peer review of "From Psychostasis to the Discovery of Cardiac Nerves: The Origins of the Modern Cardiac Neuromodulation Concept"

_biology, 2024, doi:10.3390/biology13040266_

Round 1

Reviewer 1 Report

Comments and Suggestions for Authors

This review provides a captivating history of medical and neuroanatomical research and introduces the concept of cardiac neuromodulation. It will be of great interest to biologists and researchers from other fields.

The manuscript is well-organized and thoroughly documented, includes captivating images, and is written in a simple and easy-to-understand language.

Minor comments:

The text discusses historical information that is already well-documented. However, it lacks specific citations to the documents from which this information was obtained. For instance, no citations are provided on the first page between lines 48 and 100, except for references to (1, 2, 3). Similarly, no citations are provided between lines 175 to 183 and 245 to 249. Please make sure to go through the entire manuscript carefully and check for any missing citations.

Author Response

Dear Reviewer,

Thank you for bringing the missing citations to my attention. I recognize the importance of accurate source attribution, particularly when dealing with well-documented historical information.

I have thoroughly reviewed the entire manuscript, paying specific attention to the identified sections (lines 48-100, 175-183, and 245-249). I can assure you that all historical information is now accompanied by appropriate citations, including references to the specific source documents.

Dear Reviewer, i finished to rename the references in the paper. I also added a further reference on the concept of psychostasia in Islam. Please, you can find it on line 136 (reference 11). 

Reviewer 2 Report

Comments and Suggestions for Authors

In the conclusion, the authors have written: “This comprehensive review has meticulously traced the historical trajectory of medical and neuroanatomical research on cardiac innervation, shedding light on the nascent biomolecular concepts of cardiac neuromodulation”. If this were so, the topic would certainly be of interest. Very interesting data about ancient cardiology was provided. Unfortunately, the current review appears incomplete, fragmented and superficial. The citation is clearly insufficient. Due to theses profound shortcomings, the article is not acceptable for publication in the current form. I present my concerns in no particular order of importance.

Major.

In the  abstract, there are only historical remarks. Nevertheless, in the main text the authors tried to analyze “the concept of cardiac neuromodulation” (25 of 39 pages, but this material was not reflected in the abstract). It is strange, but there is no aim in the abstract (and there is no introduction in the main text where it may be mentioned). The authors indicate the purpose only in the conclusions (see above). In the abstract and in the conclusion the authors indicated an importance of studying intrinsic and extrinsic cardiac innervation due to PASC syndrome. However, autonomic dysfunctions of heart regulation under COVID infections are not discussed anywhere in the text.

Authors gave a large volume of information about ancient science. Of course, it is quite interesting. But it is rather strange that authors focused on ancient medicine and did not mentioned Weber brothers discovered vagal inhibition, Ludwig (with Cyon) discovered the vasomotor reflexes, Stelling, Bernard, Waller and Brown-Sequard identified sympathetic as pressor nerves, Von Euler's demonstration that the sympathetic transmitter was noradrenaline etc. Nevertheless, instead of a detailed description of the history of genuine discoveries in neurocardiology, the authors provide a detailed description of «miracles». I think that such information would be interesting for the tabloid press, but not for a serious scientific review.

After information about Langley's discoveries, suddenly, in the same paragraph, a description of the neural networks of the autonomic nervous system and central control begins without any subheadings. However, these data looks like a book chapter for students with only few references.

On page 23 there is some information about ion channels in cardiac tissue. I think this is unnecessary, because the authors do not consider the functional mechanisms of the heart contraction or synaptic transmission.

When neurotransmitter phenotype is discussed in cardiac ganglia, authors mainly cited Pauza and his co-workers. However, there also many excellent studies from other groups, for example Colin Anderson from Australia (Richardson RJ, Grkovic I, Anderson CR. Immunohistochemical analysis of intracardiac ganglia of the rat heart. Cell Tissue Res. 2003 Dec;314(3):337-50. doi: 10.1007/s00441-003-0805-2). The authors focused on NO in the intracardiac ganglia, but NO is found only in a few cells in rats (Richardson et al., 2003). Besides, in rats but not in humans all intracariac neurons are NPY-IR. Why not discuss this in more detail using different references?

What about other concepts in the neurocardiology? Are the authors familiar with the work of the Armor group? (for example Ardell JL, Andresen MC, Armour JA, Billman GE, Chen PS, Foreman RD, Herring N, O'Leary DS, Sabbah HN, Schultz HD, Sunagawa K, Zucker IH. Translational neurocardiology: preclinical models and cardioneural integrative aspects. J Physiol. 2016 Jul 15;594(14):3877-909. doi: 10.1113/JP271869)?

Minor.

I would suggest avoiding single paragraph sentences. This is grammatically incorrect.

It is also not entirely correct to start all terms and anatomical names with capital letters. For example: Gamma Aminobutyric Acid (line 774), Thalamus, Hypothalamus, and Amygdala, Dorsal Motor nucleus etc

Author Response

Dear Reviewer,Thank you for taking the time to provide your valuable feedback on my article. I have carefully reviewed your comments and have prepared a detailed response, which I have attached for your convenience.In the attachment, I have addressed each of your comments and provided explanations, clarifications, and revisions where necessary. I have also highlighted (in green) the changes made to the article for your reference.I appreciate your insights and expertise, which have helped me to improve the quality and clarity of my work. I hope that the revised article now meets your expectations and I look forward to hearing your further feedback.Thank you again for your time and consideration.

Best regards

Reviewer 3 Report

Comments and Suggestions for Authors

Manuscript ID: biology-2865172

Type of manuscript: Review

Title: From Psychostasis to the Discovery of Cardiac Nerves: the Origins of the Modern Cardiac Neuromodulation Concept

Authors: Beatrice Paradiso, Dainius Pauza, Clara Limbaeck, Giulia Ottaviani, Gaetano Thiene

This review article addresses issues related to cardiac innervation. The authors present this topic based on existing literature. Knowledge encompassing contemporary research has been enriched by historical facts dating back to the times of ancient Egypt. The data presented in the manuscript are very interesting and valuable. They can serve not only as a valuable database for scientists regarding the anatomical and physiological aspects of cardiac innervation but also these pieces of information can be used for educational purposes.

However, as a reviewer, I have a few  comments for the Authors. Please refer to them and verify the manuscript.

1.         Keywords should be placed under the Abstract (Guidelines for Authors - Front matter).

2.         Abstract - In the last paragraph, the authors included information about SARS-CoV-2 infection. It's intriguing, but there is no further development of this thread in the rest of the manuscript. Please decide whether to continue and develop this topic or abandon it. It would be interesting to have discussions and suggestions on this topic, which align with the assumptions of a review article.

3.         Abbreviations - please check all abbreviations, possibly expanding them, e.g., "BP" (line 750).

- Is DMV the same as DMNV(line 390-396)? Description unclear.

-  There are a lot of abbreviations, which can be confusing. I suggest introducing a list.

4. Page 15, Fig. 6. Unclear description of the illustration. Please explain/consolidate where the illustration comes from? From Biaggioni et al or Benarroch? Expand the description under the illustration. In its current form, the description is not very clear. 5. The same situation is with Fig. 10

6. Fig 6 - I suggest including information about the colored lines (underlining the names of the nuclei) in the photo description. The sentence "The preganglionic sympathetic innervation in blue" (line 413) seems to have been "taken out of context".

7. Fig. 6 once more. What is the nature of the nucleus of the solitary tract? Parasympathetic or sensory? If the authors have data on the parasympathetic nature of this nucleus, citations should be provided in the text.

8. Lack of conclusion and indication of further potential research directions. Please complete this.

9. Please verify the manuscript with the journal's guidelines (Back matter).

10. References - please standardize the style according to the journal's guidelines.

Author Response

Dear reviewer, I revised my article following your suggestions:
1. Keywords should be placed under the Abstract (Guidelines
for Authors - Front matter). I wrote down the keywords
2. Abstract - In the last paragraph, the authors included
information about SARS-CoV-2 infection. It's intriguing, but there is
no further development of this thread in the rest of the manuscript.
Please decide whether to continue and develop this topic or
abandon it. It would be interesting to have discussions and
suggestions on this topic, which align with the assumptions of a
review article. I perfected the manuscript by delving into the problems of the dysautonomic effects of long covid
3. Abbreviations - please check all abbreviations, possibly
expanding them, e.g., "BP" (line 750). I wrote a list of abbreviations
- Is DMV the same as DMNV(line 390-396)? Description unclear.
- There are a lot of abbreviations, which can be confusing. The suggestions
introducing a list. I have combined it under a single abbreviation
4. Page 15, Fig. 6. Unclear description of the illustration. Please
explain/consolidate where the illustration comes from? From
Biaggioni et al or Benarroch? Expand the description under the
illustration. In its current form, the description is not very clear. 5.
The same situation is with Fig. 10 The figure belongs to Benarroch's text in chapter 2 of the book by Baggioni et al
6. Fig 6 - I suggest including information about the colored lines
(underlining the names of the nuclei) in the photo description. The
sentence "The preganglionic sympathetic innervation in blue" (line
413) seems to have been "taken out of context". I described the colored lines
7. Fig. 6 ounces blackberries. What is the nature of the nucleus of the solitary
tract? Parasympathetic or sensory? If the authors have data on the
parasympathetic nature of this nucleus, quotations should be provided
in the text. The nucleus of the solitary tract is an integrated ortho-parasympathetic system, including receptors of the serotonergic, dopaminergic, opioid and neuropeptide pathways. it is a reflex arc and by mistake I wanted to underline the parasympathetic afference to the nucleus of the solitary tract, but this is confusing, so I removed it
8. Lack of conclusion and indication of further potential research
directions. Please complete this. I added the discussion and conclusion chapter
9. Please verify the manuscript with the journal's guidelines (Back
matters). I tried to standardize
10. References - please standardize the style according to the
journal's guidelines. I tried to standardize

Round 2

Reviewer 2 Report

Comments and Suggestions for Authors

The authors have improved their manuscript through the first round of revision by addressing all the points reviewers raised. I recommend accepting this manuscript for the publication.